# Allosteric modulation in monomers and oligomers of a G protein-coupled receptor

Rabindra V Shivnaraine[1*†], Brendan Kelly[1‡], Krishana S Sankar[2], Dar'ya S Redka[1], Yi Rang Han[1], Fei Huang[1], Gwendolynne Elmslie[3], Daniel Pinto[1], Yuchong Li[4], Jonathan V Rocheleau[2], Claudiu C Gradinaru[4*], John Ellis[3*], James W Wells[1*]

[1]Department of Pharmaceutical Sciences, Leslie Dan Faculty of Pharmacy, University of Toronto, Toronto, Canada; [2]Department of Physiology, University of Toronto, Toronto, Canada; [3]Departments of Psychiatry and Pharmacology, Hershey Medical Center, Hershey, United States; [4]Department of Physics, University of Toronto, Toronto, Canada

*For correspondence: rvshiv@ stanford.edu (RVS); claudiu. gradinaru@utoronto.ca (CCG); JohnEllis@psu.edu (JE); jwells@ phm.utoronto.ca (JWW)

Present address: †Department of Molecular and Cellular Physiology, Stanford University School of Medicine, Stanford, United States; ‡Department of Computer Science, Stanford University, Stanford, United States

Competing interests: The authors declare that no competing interests exist.

**Abstract** The $M_2$ muscarinic receptor is the prototypic model of allostery in GPCRs, yet the molecular and the supramolecular determinants of such effects are unknown. Monomers and oligomers of the $M_2$ muscarinic receptor therefore have been compared to identify those allosteric properties that are gained in oligomers. Allosteric interactions were monitored by means of a FRET-based sensor of conformation at the allosteric site and in pharmacological assays involving mutants engineered to preclude intramolecular effects. Electrostatic, steric, and conformational determinants of allostery at the atomic level were examined in molecular dynamics simulations. Allosteric effects in monomers were exclusively negative and derived primarily from intramolecular electrostatic repulsion between the allosteric and orthosteric ligands. Allosteric effects in oligomers could be positive or negative, depending upon the allosteric-orthosteric pair, and they arose from interactions within and between the constituent protomers. The complex behavior of oligomers is characteristic of muscarinic receptors in myocardial preparations.

## Introduction

Muscarinic acetylcholine receptors contain an orthosteric site and a topographically distinct allosteric site (*Kruse et al., 2014*; *May et al., 2007*). The latter is located at the extracellular surface within a vestibule to the orthosteric site (*Kruse et al., 2013*), and that region shows comparatively low sequence homology among the five muscarinic subtypes ($M_1$–$M_5$). The allosteric site therefore is more subtype-specific than the orthosteric site, and allosteric ligands are increasingly of interest for their therapeutic potential (*Kenakin2004*). Owing to the early demonstration of allosteric interactions in muscarinic systems (*Clark et al., 1976*; *Stockton et al., 1983*) and the variety of available modulators (*May et al., 2007*), the $M_2$ muscarinic receptor has been a prototype for such effects within the broader family of G protein-coupled receptors (GPCRs).

Most studies of the interaction of an allosteric ligand with the allosteric site have been based on the modulation of events at the orthosteric site, typically on receptors in native membranes or detergent-solubilized extracts. Such events may be measured directly, as in the binding of a radiolabeled antagonist (*Christopoulos et al., 2002*), or they may be inferred from functional consequences such as the turnover of [$^{35}$S]GTPγS (*May et al., 2007*). They also may be positive or negative (*May et al., 2007*), and the data generally have been described in terms of interactions between two binding sites on a monomeric receptor (*Christopoulos et al., 2002*).

**eLife digest** Proteins called G protein-coupled receptors (GPCRs) are found on the surface of cells throughout the body. Hormones or other signal molecules – collectively known as ligands – from outside the cell can bind to the receptors to activate them. This causes a change in the structure of the receptor, which triggers a signal inside the cell to alter the cell's behavior. GPCRs are known to form clusters of two or more receptor units, but it is not known if these clusters have unique properties or what role they play in cells.

Many drugs can bind to GPCRs and most of them block the activity of the receptors by taking the place of the natural ligand. Another way to alter the activity of a GPCR is with so-called 'allosteric' drugs. These bind to different sites on the receptor than the natural ligands do and can inhibit or enhance binding of the ligands by altering the shape of the receptor.

Shivnaraine et al. investigated how a type of GPCR called muscarinic cholinergic receptors interact within clusters. This involved developing a method to track the receptor in mammalian cells using a fluorescent sensor that detects changes in the allosteric site. The experiments show that two or more GPCRs need to interact for the receptors to respond to allosteric drugs in a manner that reflects the normal effect of the drugs on the body. This result is unexpected in light of the assumption that individual receptor molecules act independently. Shivnaraine et al.'s findings indicate that the clusters may play a role in the normal behavior of GPCRs in cells. A future challenge is to understand exactly how the GPCRs interact with each other.

In the case of the $M_2$ receptor, an atomic-level view of the interaction within a monomer has emerged recently from the crystallography-derived structure of a biliganded receptor (*Kruse et al., 2013*) and from inferences based on molecular dynamics simulations (*Dror et al., 2013*). Although such results offer an explanation for negative cooperativity at the atomic level, the mechanism of action of positive modulators such as strychnine remains unclear (*Dror et al., 2013*). That uncertainty relates to questions regarding electrostatic repulsion between highly charged allosteric and orthosteric ligands, steric effects of the former on binding of the latter, and the effect of either ligand on conformational stability at the site of the other (*Dror et al., 2013*).

The common view that allosteric interactions occur within monomeric receptors is limited in its ability to rationalize complex effects that are seen in the binding of radioligands. For example, dissociation of the antagonist [$^3$H]quinuclidinylbenzilate (QNB) from the $M_2$ receptor is accelerated by gallamine at lower concentrations of the allosteric ligand and slowed at higher concentrations, resulting in a bell-shaped profile (*Ellis et al., 1989*). At equilibrium, the binding of [$^3$H]QNB and the inverse agonist *N*-[$^3$H]methylscopolamine (NMS) can display a biphasic and even a triphasic dependence on the concentration of an allosteric modulator (*Proska et al., 1994*; *Shivnaraine et al., 2012*). Such effects imply either that a single molecule of the receptor possesses up to four allosteric sites or that allostery occurs via linked sites within an oligomer (*Proska et al., 1994*; *Shivnaraine et al., 2012*).

To understand how a multimeric complex might account for allosteric behavior that resists explanation in terms of monomers, we have compared positive and negative allosteric modulators for their effects on oligomers and purified monomers of the $M_2$ muscarinic receptor. Our approach has involved a novel FRET-based sensor of conformation at the allosteric site, mutants that allow only for allosteric modulation between linked protomers, mechanistic modeling, and molecular dynamics simulations. Taken together, the results indicate that intramolecular interactions—*i.e.*, between two sites on a monomer or on the same protomer of an oligomer—are dominated by electrostatic repulsion and result in low-affinity negative modulation by the allosteric ligand. Intermolecular interactions—*i.e.*, between two sites on neighboring protomers of an oligomer—result in high-affinity allosteric modulation that may be positive or negative, depending upon the constraints associated with ligand-binding and the nature and extent of conformational changes transmitted between protomers. The results provide a direct demonstration of how allosteric effects characteristic of $M_2$ receptors arise from interactions between the constituent protomers of an oligomer.

## Results

### Binding of allosteric modulators to monomers

$M_2$ receptors were solubilized as oligomers and purified as monomers in the manner described previously (*Redka et al., 2013*; *2014*). cMyc- and FLAG-tagged receptors were extracted from co-infected *Sf9* cells in digitonin–cholate, and oligomers were detected by co-immunoprecipitation (*Figure 1A*). The solubilized preparation was applied to an affinity resin of immobilized aminobenztropane (ABT), and western blotting of the purified receptor with an anti-$M_2$ antibody indicated that 97% of the immunopositive material migrated as a monomer. Eighty-three percent migrated as a monomer when the sample was cross-linked with $BS^3$ (*Figure 1B* and *Figure 1—source data 1*). The affinity of [$^3$H]NMS for the purified monomer (*Equation 3*, log $K = -8.01 \pm 0.04$, $N = 5$) and the purity of the sample were the same as reported previously (cf. *Redka et al., 2013*; *2014*).

Monomeric $M_2$ receptors retain the interaction between ligands at the allosteric and orthosteric sites. The allosteric modulator gallamine slowed the dissociation of [$^3$H]QNB, and the time-course of the dissociation was mono-exponential under all conditions. The dose-dependence of the decrease in the rate constant was monophasic with a Hill coefficient of 1 (*Figure 1C*, *Figure 1—source data 1*). Gallamine therefore appears to slow the release of [$^3$H]QNB from monomers via a single allosteric site. In contrast, two or more allosteric sites can be inferred from the bell-shaped behavior of receptors that are predominantly or wholly oligomeric (*Shivnaraine et al., 2012*) (*Figure 1C*, *Figure 1—source data 1*).

Monomers of the $M_2$ receptor equilibrated slowly with [$^3$H]NMS and either gallamine or the allosteric modulator strychnine when the two ligands were added simultaneously, and incubation for 21 hr was required for the attainment of equilibrium at 30°C. The binding profile upon equilibration was monophasic downward in each case, and the data can be described by a single hyperbolic term (*i.e.*, *Equation 2*, $n = 1$) (*Figure 1D and E*; *Figure 1—source data 1*). Gallamine and strychnine therefore appear to modulate the equilibrium binding of [$^3$H]NMS to monomers via a single allosteric site.

The monophasic nature of effects observed at purified monomers differs from the triphasic effect of gallamine on the binding of [$^3$H]NMS to $M_2$ receptors in other preparations, including membranes and detergent-solubilized extracts from *Sf9* cells, CHO cells, and porcine atria (*e.g.*, *Figure 1D*, broken line) (*Shivnaraine et al., 2012*). Those multiphasic curves and the Hill coefficients of the individual components, taken together, are indicative of at least four interacting allosteric sites, which in turn are suggestive of four interacting receptors within a tetramer or larger oligomer. Similarly, the bell-shaped pattern obtained for strychnine in atrial extracts (*Figure 1E–G*) and in membranes from CHO cells (*Figure 1—figure supplement 1*, *Figure 1—source data 2*) is indicative of at least two allosteric sites and also points to an oligomer.

Allosteric ligands bind in a vestibule to the orthosteric site (*Kruse et al., 2013*), forming a cap that impedes the binding and dissociation of the orthosteric ligand (*Figure 6*) (*Shivnaraine et al., 2012*). Interference by one ligand in the binding kinetics of another leads to binding patterns that depend upon the order of mixing in a system that has not attained equilibrium. Such kinetic effects were examined experimentally by varying the sequence in which strychnine and [$^3$H]NMS were added to the receptor. In one protocol, the two ligands were added simultaneously and incubated with the receptor for 21 hr to obtain the pattern at equilibrium. In another, one ligand was pre-equilibrated with the receptor prior to the addition of the second, and the mixture was incubated for a further 3 hr to obtain the pattern at a time prior to equilibrium.

With receptors in atrial extracts, the effect of strychnine on the binding of [$^3$H]NMS was bell-shaped under all conditions (*Figure 1E–G*, *Figure 1—source data 1*). With purified monomers, strychnine was strictly inhibitory except when the receptor was pre-equilibrated with [$^3$H]NMS, when there was no effect (*Figure 1E–G*, *Figure 1—source data 1*). The patterns observed in experiments on monomers are consistent with the patterns observed in simulations performed according to *Figure 6* (*Figure 1—figure supplement 2*, *Figure 1—source data 3*), which describes a mechanism for allosteric capping of the orthosteric site in a monomer.

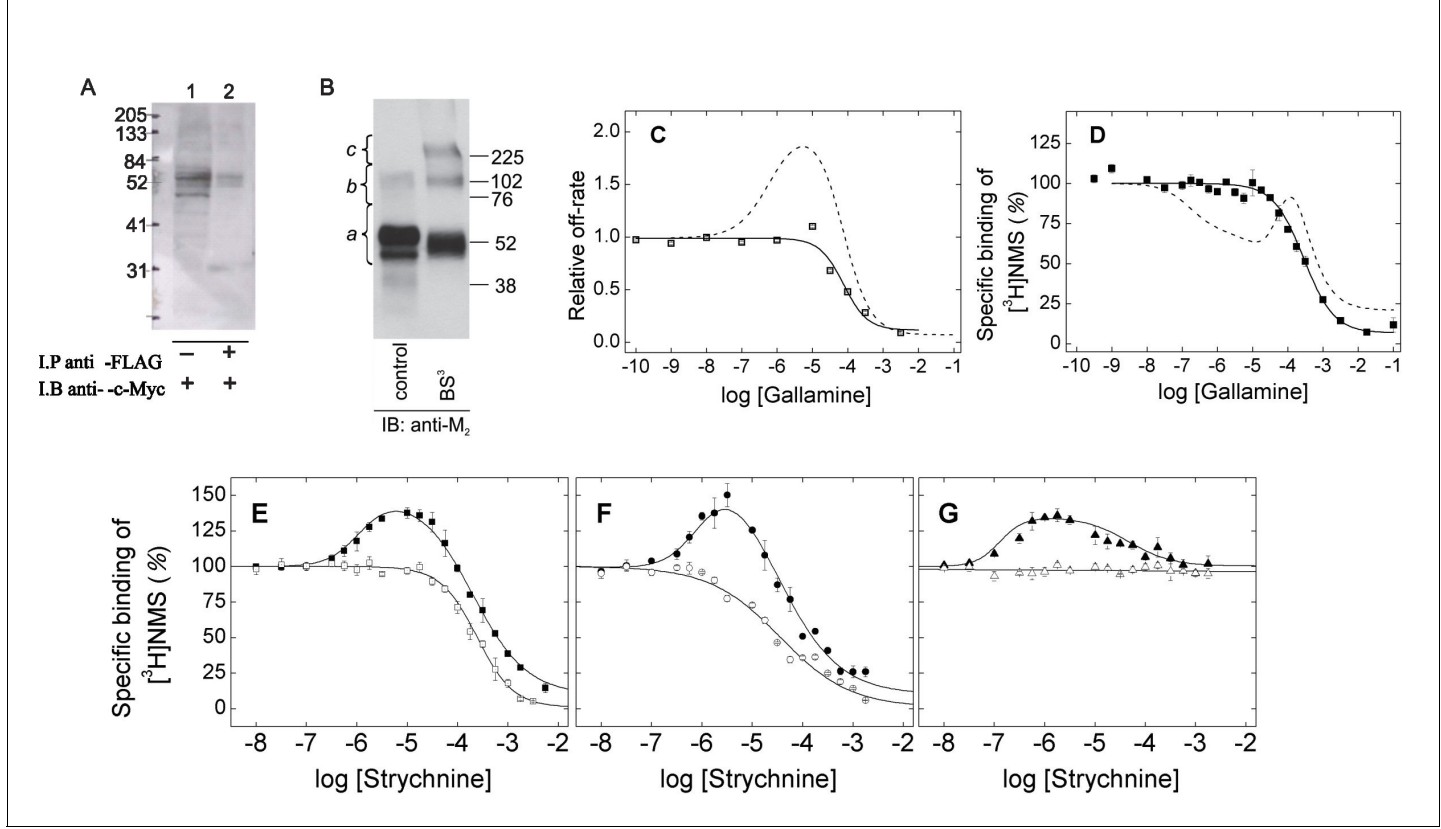

**Figure 1.** Oligomeric and monomeric preparations of the $M_2$ muscarinic receptor. (**A**) Gels were loaded with samples of FLAG- and c-Myc-tagged receptor extracted from co-infected *Sf9* cells (lane 1, 20–30 fmol of receptor per lane) or from the precipitate obtained upon treatment of the extract with an immobilized anti-FLAG antibody (lane 2, 15–35 fmol of receptor per lane). The amount of receptor was determined from the binding of [³H]QNB at the saturating concentration of 10 nM. Following electrophoresis and transfer, the membranes were blotted with anti-c-Myc antibody. (**B**) Gels were loaded with parallel samples of purified $M_2$ receptor taken before (lane 1) and after cross-linking with BS³ (lane 2). The same amount of receptor was applied to each well (15–25 fmol). The intensities of the immunopositive bands were measured by densitometry, and the area under each densitometric trace was determined in three segments as indicated by the braces (*a*, 40–75 kDa; *b*, 75–170 kDa; *c*, 170–360 kDa) (*Figure 1—source data 1*, N = 3). The monomeric receptor migrated as a doublet (~44 kDa and 53 kDa). (**C**) The rate constant for the dissociation of [³H]QNB from purified monomers was measured at graded concentrations of gallamine ($k_{obsd}$) and normalized to that in the absence of gallamine ($k_0$) to obtain the relative rate constant ($k_{obsd}/k_0$) plotted on the *y*-axis. The data were analyzed in terms of *Equation 2* to obtain the fitted curve shown in the figure (solid line) and the parametric values listed in *Figure 1—source data 1*. (**D**) [³H]NMS (10 nM) was mixed with gallamine at the concentrations shown on the *x*-axis, and binding was measured after incubation of the reaction mixture for 21 hr at 30°C. The solid line represents the best fit of *Equation 2* (n = 1), and the fitted parametric values are listed in *Figure 1—source data 1*. The dashed lines in panels **C** and **D** are the fitted curves from similar experiments on preparations in which the $M_2$ receptor is largely or wholly oligomeric (**C**, *Sf9* membranes; **D**, *Sf9* extracts) (*Shivnaraine et al., 2012*). (**E–G**) The binding of [³H]NMS (10 nM) to $M_2$ receptors extracted from porcine sarcolemmal membranes (**closed symbols**) and purified as monomers from *Sf9* cells (**open symbols**) was measured at graded concentrations of strychnine following the simultaneous addition of both ligands (**E**), the sequential addition of strychnine and [³H]NMS (**F**), and the sequential addition of [³H]NMS and strychnine (**G**). The binding profiles obtained following simultaneous addition were identical after incubation of the samples for 3 hr and 21 hr, and the data obtained after 21 hr are shown in the figure. Pretreatment with one ligand was followed after 2 hr by the addition of the other and further incubation for 3 hr. The temperature of incubation was 30°C throughout. The solid lines in each panel depict the best fits of *Equation 2* to the data from three experiments taken in concert, and the parametric values are listed in *Figure 1—source data 1*.

The following source data and figure supplements are available for figure 1:

**Source data 1.** *Panel B*–Monomeric status of the purified $M_2$ receptor after chemical cross-linking.

**Source data 2.** Figure 1–figure supplement 1–Parametric values for the effect of strychnine on the binding of [³H]NMS and [³H]QNB to membrane-bound $M_2$ receptor.

**Source data 3.** Figure 1-figure supplement 2–Rate constants for the simulated binding of strychnine and [³H]NMS according to *Figure 6*.

*Figure 1 continued on next page*

*Figure 1 continued*

**Figure supplement 1.** Effect of strychnine on the binding of [³H]NMS and [³H]QNB to membrane-bound $M_2$receptor.

**Figure supplement 2.** Simulated effect of strychnine on the binding of [³H]NMS according to *Figure 6*.

## A FRET-based sensor of allosteric effects

Allosteric effects that produce multiphasic binding profiles such as those illustrated by the dashed lines in *Figure 1C,D* have been attributed to a combination of inter- and intramolecular interactions within oligomers (*Shivnaraine et al., 2012*). For more direct information on those interactions and related conformational changes, we developed a sensor based on FRET between FlAsH and mCherry. FlAsH was incorporated via the hairpin-forming sequence FLNCCPGCCMEP (FCM) (*Hoffmann et al., 2010*), which was inserted after Val[166] in the second extracellular loop (ECL2), and mCherry was fused to the amino terminus (*i.e.*, mCh-$M_2$-FCM).

A fluorophore at the *N*-terminus of the $M_2$ receptor has been shown previously to have no discernible effect on various properties, including transport of the receptor to the plasma membrane, agonist-induced responses, and the binding of muscarinic ligands (*Pisterzi et al., 2010*). Localization at the plasma membrane also was not affected by the modifications described here, as visualized by fluorescence from either the fused fluorophore or bound FlAsH (*Figure 2—figure supplement 1*), nor was there any apparent change in the binding properties. The affinity of the receptor for [³H] NMS was essentially the same, as measured in preparations of membrane-bound and detergent-solubilized mCh-$M_2$-FCM from transfected CHO cells (*Equation 3*: membranes, log $K = -9.52 \pm 0.10$, $N = 3$; extract, log $K = -8.26 \pm 0.22$, $N = 3$). The modified receptor also retained the triphasic allosteric effect of gallamine on the binding of [³H]NMS, as measured in membranes from CHO cells expressing mCh-$M_2$-FCM (*Figure 2—figure supplement 2*).

Molecular models of the native $M_2$ receptor, mCh-$M_2$-FCM, and the FlAsH-bound sensor were rendered from the crystal structures of mCherry (2H5Q) (*Shu et al., 2006*) and the receptor in an active state (4MQS) (*Kruse et al., 2013*). A structure of the sensor showing the positions of FlAsH and mCherry is given in *Figure 2A*. Insertion of the FlAsH-reactive sequence in ECL2 created a rigid extended loop (*Figure 2B*) without perturbing the conformation of the loop in the region of the adjacent EDGE motif. There was no distortion of the extended loop upon the addition of FlAsH.

To examine the mobility of a fluorophore at the amino terminus, eGFP was fused to the wild-type $M_2$ receptor (eGFP-$M_2$) and to a truncated mutant lacking the first 13 amino acids (eGFP-trunc$M_2$). This truncation shortened the link between the fluorophore and the first helical domain and was expected to increase the likelihood that the movement of eGFP would approximate that of the fusion protein as a whole. Each tagged receptor was expressed in CHO cells and extracted in digitonin–cholate, and the rotational correlation time ($\varphi$) of the fused fluorophore was estimated from the fluorescence anisotropy according to *Equations 4–9*. The full-length receptor was examined with and without ligands, and eGFP alone was taken as a control.

Both the fluorescence (*Equation 6*) and the anisotropy (*Equation 9*) decayed as a single exponential under all conditions, and the parametric values are listed in *Figure 2—source data 1*. The value of 18 ns obtained for the correlation time of free eGFP agrees favourably with that of about 20 ns reported previously (*Devauges et al., 2012*) and was sixfold longer than the fluorescence lifetime of 3 ns. In contrast, values of 30–55 ns were obtained for eGFP-trunc$M_2$, eGFP-$M_2$, and the various liganded states of eGFP-$M_2$. Such values are at least 12–22-fold longer than the corresponding lifetimes of 2.5–2.6 ns, and the latter are much shorter than the 12-ns measuring window of the fluorescence decay. The rotational correlation times obtained for receptor-bound eGFP therefore are indistinguishable, and the movement of the fluorophore at the *N*-terminus of the receptor appears to be determined primarily by the tumbling of the whole protein, with or without ligands.

Excitation of bound FlAsH at 498 nm gave the observed emission spectrum and unmixed components shown for a typical cell in *Figure 2C*. Fluorescence from the acceptor accounted for most of the total signal at 610 nm, and the corresponding peak derived predominantly from FRET. Only 6% of the emission from the acceptor came from direct excitation, and it was accounted for during spectral unmixing. The apparent FRET efficiency of each cell was calculated from the normalized amplitudes of the unmixed spectra (*Equations 10 and 12*), and the efficiencies from all cells gave a

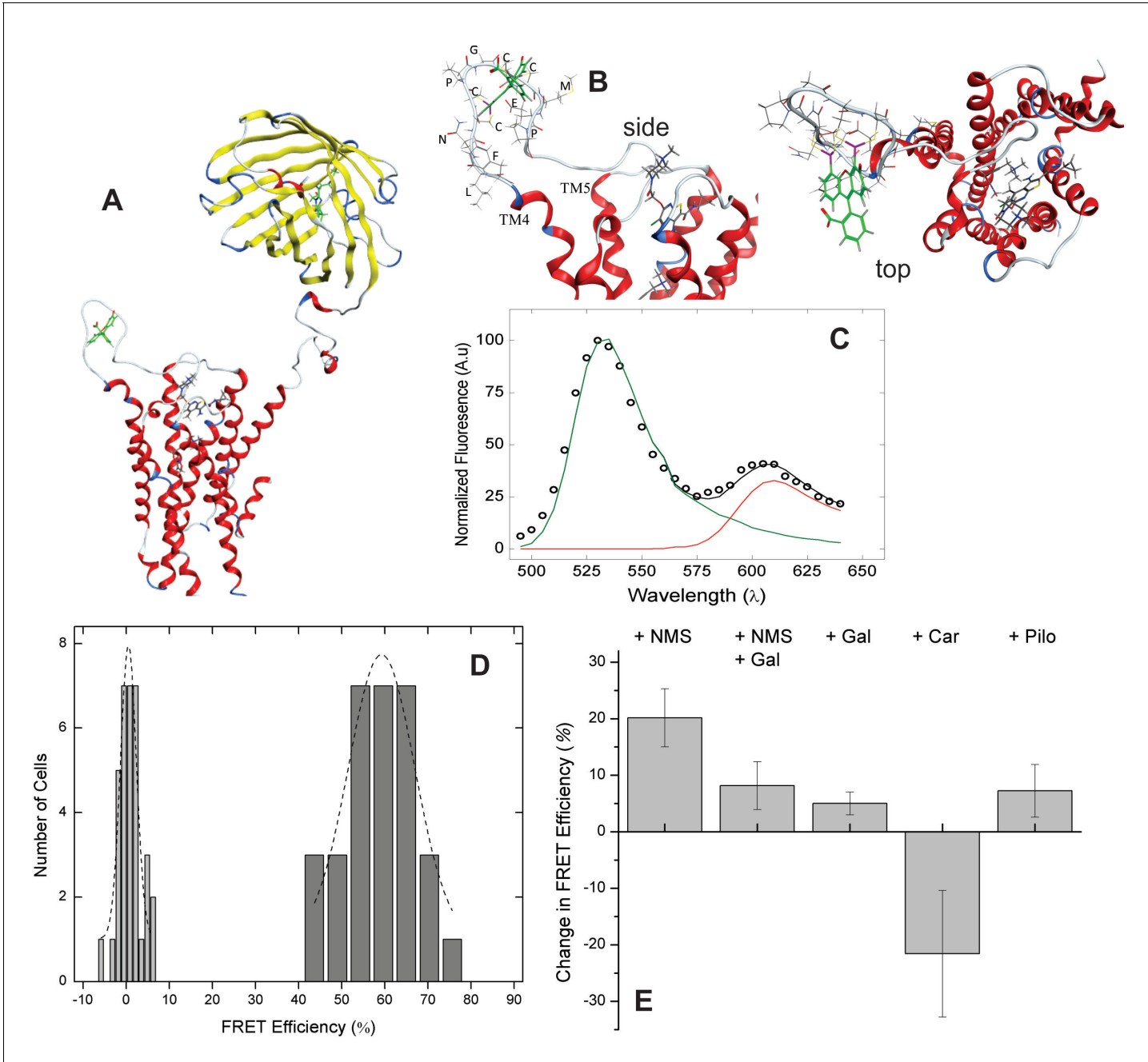

**Figure 2.** FRET-based detection of conformational change at the allosteric site. (**A**) A computed molecular model of the $M_2$ receptor (red) fused at the N-terminus to mCherry (yellow) and labeled with FlAsH (green) at a tetracysteine motif inserted in ECL2 between Val[166] and Gly[167]. (**B**) Expanded views of ECL2 with the insert FCM and FlAsH. (**C**) $M_2$ receptor bearing mCherry and the FlAsH-reactive insert (mCh-$M_2$-FCM) was expressed in CHO cells and treated with FlAsH. Images were collected upon excitation at 498 nm and 0.37 μW. The emission spectrum from a single cell is shown in the figure (**C**). The spectrum was unmixed (**Equation 10**) to obtain the contributions of donor ($k_D$, green) and acceptor ($k_A$, red) to the fitted sum (black). (**D**) CHO cells expressing mCh-$M_2$-FCM or the mCherry-tagged wild-type $M_2$ receptor (mCh-$M_2$) were treated with FlAsH and excited at 498 nm. The FRET efficiencies ($E_{app}$) calculated for individual cells are shown as histograms, and the dashed lines depict the best fits of the Gaussian distribution (dark grey bars, mCh-$M_2$-FCM, 31 cells, $\mu$ = 59.2 ± 1.2, $\sigma$ = 7.7 ± 3.0; light grey bars, mCh-$M_2$, 34 cells, $\mu$ = 0.50 ± 0.35, $\sigma$ = 1.8 ± 0.5. (**E**) mCh-$M_2$-FCM was expressed in CHO cells and reacted with FlAsH, and the value of $E_{app}$ was measured before and after addition of the inverse agonist NMS (1 μM, N = 26), the allosteric modulator gallamine (Gal, 10 mM, N = 42), NMS (1 μM) plus gallamine (10 mM, N = 18), the agonist carbachol (Car, 1 mM, N = 19), and the partial agonist pilocarpine (Pilo, 1 mM, N = 19). The ligand-dependent changes in $E_{app}$ are plotted in the figure (*i.e.*, $\Delta E_{app}$ ± S.D.), and the values are listed and compared in **Figure 2—source data 1**. The mean value of $E_{app}$ for cells in the absence of ligand was 60 ± 8% (N = 26).

*Figure 2 continued on next page*

*Figure 2 continued*

The following source data and figure supplements are available for figure 2:

**Source data 1.** *Panel A–Time-resolved fluorescence and fluorescence anistorary of eGFP and eGFP-tagged M$_2$ receptors.*

**Figure supplement 1.** Expression and localization of fluorophore-tagged M$_2$ receptors.

**Figure supplement 2.** Modulatory effect of gallamine on the binding of [$^3$H]NMS to mCh-M$_2$-FCM.

**Figure supplement 3.** FRET between eGFP fused at the *N*-terminus and mCherry inserted after Val$^{166}$.

distribution centered on 60% with a width of 18% at half-maximal amplitude (*Figure 2D*). Controls in which FlAsH was reacted with mCherry-tagged receptors lacking the FCM sequence gave a narrow distribution centered on 0.5% (*Figure 2D*), confirming the specificity of the reaction with FlAsH.

The emission spectrum and corresponding FRET efficiency between FlAsH and mCherry was measured for each cell before and after the addition of various ligands. The inverse agonist NMS caused a marked increase in the amplitude of the peak near 610 nm, as illustrated in *Figure 3A*. Smaller increases were obtained with the partial agonist pilocarpine, the allosteric modulator gallamine, and the combination of gallamine plus NMS. In contrast, the agonist carbachol caused a decrease. The differences in efficiency at individual cells were averaged over all cells to obtain the mean for each ligand (*Figure 2E*, *Figure 2—source data 1*), which varied from an increase of 20 percentage points with NMS to a decrease of 22 percentage points with carbachol. The partial agonist pilocarpine caused an increase of 7.3 percentage points. These changes indicate that the sensor can distinguish between agonists and an inverse agonist at the orthosteric site. Gallamine caused an increase of 5.0 percentage points and reduced the increase caused by NMS from 22 to 8.0 percentage points, indicating that the sensor detects the effect of the allosteric ligand on NMS.

A receptor bearing eGFP at the *N*-terminus and mCherry rather than FlAsH-FCM after Val$^{166}$ in ECL2 gave a comparatively narrow distribution of FRET efficiencies with a mean of about 10% and a width of about 4% (*Figure 2—figure supplement 3A*). The effects of NMS and gallamine on FRET were eliminated (*Figure 2—figure supplement 3B*), suggesting that the insertion of mCherry within ECL2 disrupts the vestibule to the orthosteric site. The distribution of FRET efficiencies was narrower than that obtained with FlAsH and mCherry in the sensor, where the greater width of 40% may arise from differences in the labelling efficiency of FlAsH among different cells.

## Intermolecular modulation of FRET

The substitution of alanine for aspartic acid at position 103 of the M$_2$ receptor removes the counterion for positively charged ligands at the orthosteric site (*Haga et al., 2012*). That mutation has been shown previously to prevent the specific binding of [$^3$H]NMS or [$^3$H]QNB to M$_2$ receptors in HEK293 membranes at concentrations of the radioligand up to 10 nM (*Heitz et al., 1999*). The same mutation in the sensor prevented the specific binding of [$^3$H]NMS at a concentration of 1 μM, as measured with mCh-M$_2$(D103A)-FCM extracted from transfected CHO cells in digitonin–cholate. It also eliminated the effect of NMS on FRET. There was essentially no change in the emission spectrum at 610 nm (*Figure 3B*), in contrast to the increase observed with FlAsH-treated mCh-M$_2$-FCM (*Figure 3A*); also, the NMS-dependent increase in the mean FRET efficiency was reduced from 21.6 ± 11.2 percentage points in mCh-M$_2$-FCM to 0.45 ± 0.55 percentage points in the mutant (*Figure 3D*, *Figure 3—source data 1*).

Sensitivity to NMS was recovered when the binding-deficient mutant was co-expressed with the wild-type M$_2$ receptor (*Figure 3C and D*), resulting in a ligand-dependent change in the mean FRET efficiency of 9.5 ± 1.8 percentage points (*Figure 3D*, *Figure 3—source data 1*). The recovery indicates that M$_2$ receptors occur at least partly as oligomers in which the conformation in the region of the allosteric site of one protomer is affected by a ligand at the orthosteric site of another.

In cells such as that represented in *Figure 3C*, co-existence of the wild-type receptor and FlAsH-reacted mCh-M$_2$(D103A)-FCM is inferred from the effect of NMS on FRET. To confirm the cellular co-localization of the two proteins, cells were co-transfected with the plasmids for mCh-M$_2$(D103A)-FCM and the wild-type receptor fused at the *N*-terminus to eGFP (eGFP-M$_2$). Those FlAsH-treated

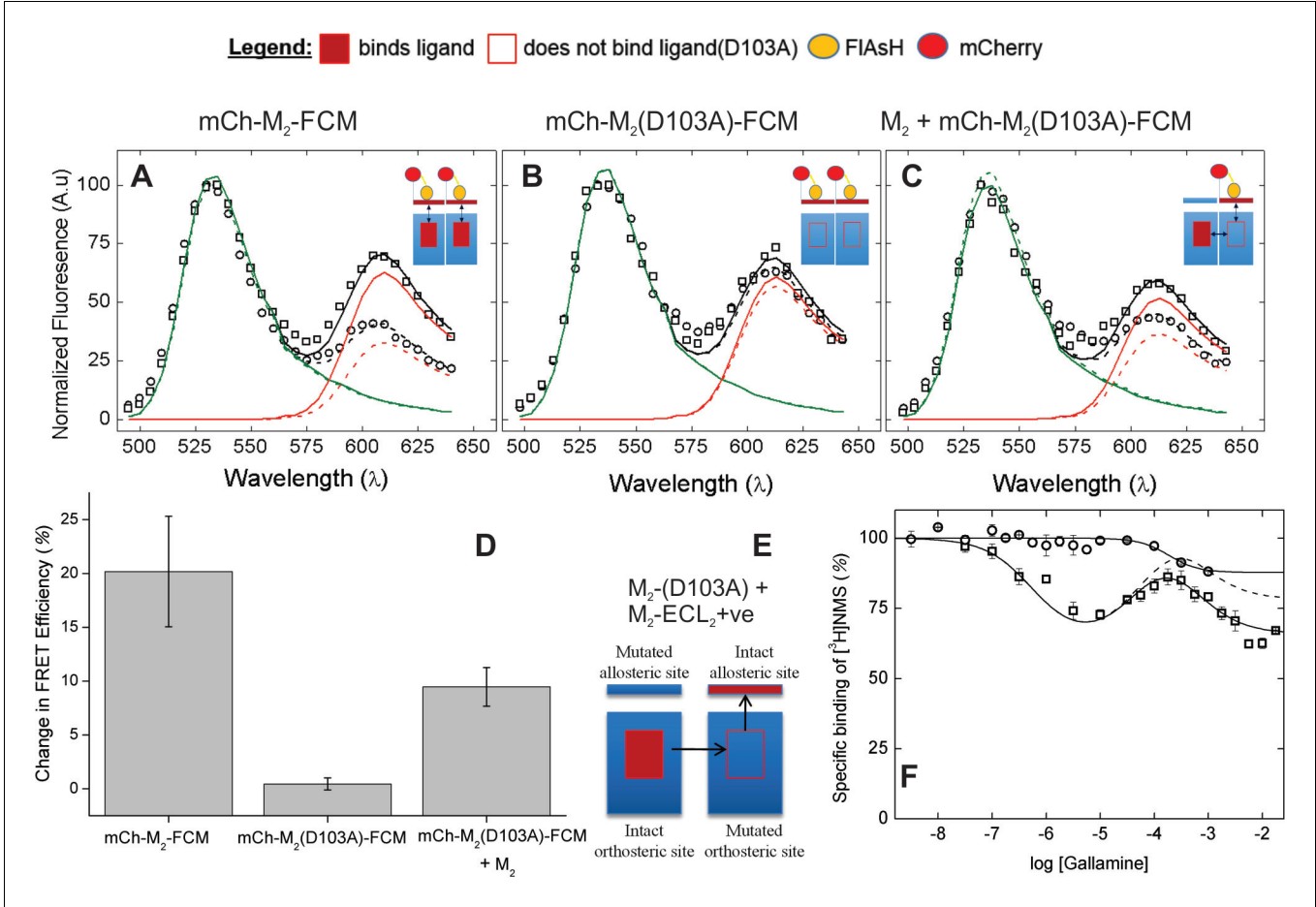

**Figure 3.** The nature of high-affinity allosteric interactions. The sensor (mCh-$M_2$-FCM) (**A**), a mutant thereof that does not bind NMS [mCh-$M_2$(D103A)-FCM] (**B**), and the mutant plus the wild-type $M_2$ receptor (**C**) were expressed or co-expressed in CHO cells and reacted with FlAsH. The cultures were irradiated before (– – ∘ – –) and after (— □ —) the addition of NMS (1 µM), and the emission spectra from individual cells were unmixed to determine the contribution of each component ($k_D$ and $k_A$, **Equation 10**) and the corresponding FRET efficiency ($E_{app}$, **Equation 12**). (**D**) Ligand-dependent changes in the FRET efficiency ($\Delta E_{app}$) were averaged over 18–42 cells transfected as described for panels **A–C**, and the means (± S.D.) are plotted in the figure. The values are listed and compared in **Figure 3—source data 1**. The value for mCh-$M_2$-FCM is replotted from **Figure 2F**. (**E**) A depiction of intermolecular cooperativity within a hetero-oligomer of the $M_2$ receptor in which binding is precluded at the orthosteric site of one mutant ($M_2$-D103A) and the allosteric site of another ($M_2$-ECL$_2$+ve). (**F**) CHO cells expressing $M_2$-ECL2+ve alone (∘) or together with $M_2$-D103A (□) were solubilized in digitonin–cholate, and hetero-oligomers were purified as described in the text. Aliquots of each sample were added to solutions of [³H]NMS (10 nM) and gallamine at the concentrations shown on the x-axis, and binding was measured after incubation of the mixture for 21 hr at 30°C. The solid lines represent the best fits of **Equation 2** to the combined data from 3 experiments, and the parametric values are as follows: ∘, $n = 1$, log $K = -3.46 \pm 0.16$; □, $n = 3$, log $K_1 = 6.25 \pm 0.16$, log $K_2 = 4.29 \pm 0.37$, log $K_3 = 3.31 \pm 0.29$. All values of $n_{H(i)}$ were indistinguishable 1 and fixed accordingly (p≥0.06). The broken line is the difference between the fitted curves for $M_2$-ECL2+ve and the copurified heteromer. [³H]NMS bound to detergent-solubilized $M_2$-ECL2+ve with an affinity of 5.5 nM (**Equation 3**, log $K = -8.26 \pm 0.10$, $n_H = 1$, $N = 3$).

The following source data and figure supplement are available for figure 3:

**Source data 1.** *Panel D—Levels of significance for ligand-dependent changes in the FRET efficiency of FlAsH-reacted mCh-$M_2$-FCM and mCh-$M_2$(D103A)-FCM*

**Figure supplement 1.** Coexpression of the $M_2$ receptor and a non-binding sensor.

cells displaying three fluorophores, and therefore both proteins, were identified by their spectral properties, and those spectra were unmixed to obtain the individual contribution from each fluorophore (**Figure 3—figure supplement 1**). NMS increased the FRET efficiency between FlAsH and mCherry by 5.3 ± 0.1 percentage points ($N = 26$) rather than 9.5 percentage points, in a further

indication that the conformational change detected by FRET in one protomer derives from the binding of the ligand to another.

## Intermolecular modulation of binding

The first two inflections of the triphasic binding profile exhibited by gallamine have been attributed to intermolecular modulation via allosteric sites on different protomers of an oligomer (*Shivnaraine et al., 2012*). In a direct test for such interactions, the three negatively charged residues of the EDGE motif in ECL2 were replaced by three positively charged residues (*i.e.*, KRGK) to obtain a mutant in which the binding of a cationic ligand such as gallamine to the allosteric site is precluded by electrostatic repulsion ($M_2$-ECL2+ve). When the mutant was expressed in CHO cells and extracted in digitonin–cholate, the substitution was found to be without effect on the affinity of [$^3$H]NMS for the orthosteric site (*Equation 3*, log $K$ = −8.26 ± 0.10, $N$ = 3). The triphasic pattern revealed by gallamine at the wild-type receptor was lost, however, and in its place was a single inflection of comparatively low affinity (log $K$ = −3.46 ± 0.16) (*Figure 3F*).

FLAG-tagged receptors lacking the orthosteric site (FLAG-$M_2$(D103A)) then were co-expressed with hexahistidyl-tagged receptors lacking the allosteric site (His$_6$-$M_2$-ECL2+ve), and purified complexes containing at least one copy of each mutant were obtained by successive passage of detergent-solubilized material on an immuno-affinity column of immobilized anti-FLAG antibody (Santa Cruz Biotechnology) and a chelating resin of $Ni^{2+}$-NTA (*Figure 3E*). Binding of [$^3$H]NMS to the purified heteromer displayed a triphasic dependence on the concentration of gallamine (*Figure 3F*), and the apparent affinities calculated in terms of *Equation 2* (*Figure 3F*, legend) were in good agreement with those reported for the wild-type receptor (*Figure 1—source data 1*) (*Shivnaraine et al., 2012*). The magnitude of the inhibitory effect corresponding to the allosteric sites of weakest affinity was much less than that observed with the wild-type receptor (cf. *Figure 1D*, broken line). It was diminished further when corrected by subtraction of the single inhibitory component observed with $M_2$-ECL2+ve (*Figure 3F*, broken line).

## Molecular dynamics simulations of the liganded receptor

Allosteric effects were examined in molecular dynamics simulations at the atomic level, starting from the crystal structure of the monomeric $M_2$ receptor occupied by the agonist iperoxo (4MQS) (*Kruse et al., 2013*). The ligand and accessory proteins were removed, and conformations of the receptor were sampled from a canonical ensemble accessible within a period of 30 ns. Simulations were performed in the absence of membrane and detergent, and ligands were docked prior to initiating the calculation. Interactions between allosteric and orthosteric ligands (*Figure 4—figure supplement 1*) were found to be affected by the degree of electrostatic repulsion between their positive charges and the conformational changes induced in one site by a ligand at the other, which together determine the positional stability of each ligand in its respective site.

The effect of electrostatic repulsion was evaluated by two correlates (*Table 1*): the distance between the spatial centers of the cationic nitrogen atoms of the two ligands (*e.g.*, strychnine and NMS, *Figure 4A*; gallamine and QNB, *Figure 4B*), and changes in the electrostatic potential of an allosteric ligand upon the addition of an orthosteric ligand. All four allosteric–orthosteric pairs showed a degree of electrostatic repulsion, but steric effects reduced the difference between strychnine and gallamine to less than would be predicted on the basis of charge alone (*Table 1*). Gallamine

**Table 1.** Correlates of electrostatic repulsion between orthosteric and allosteric ligands. The inter-cationic distance was calculated as that between the cationic nitrogen atom of the orthosteric ligand and the closest cationic nitrogen atom of the allosteric ligand. The difference in electrostatic potential was calculated as the increase in electrostatic energy of an allosteric modulator at a receptor with a vacant orthosteric site over that of the same modulator at a receptor with NMS or QNB at the orthosteric site.

| | Allosteric–orthosteric pair | | | |
| --- | --- | --- | --- | --- |
| | Str_R_NMS[a] | Str_R_QNB | Gal_R_NMS | Gal_R_QNB[a] |
| Inter-cationic distance (Å) | 13.7 | 15.7 | 16.5 | 16.8 |
| Difference in electrostatic potential (kcal/mol) | 0.0085 | 0.063 | 0.096 | 0.41 |

[a] The distances are shown in *Figures 4A and B*.

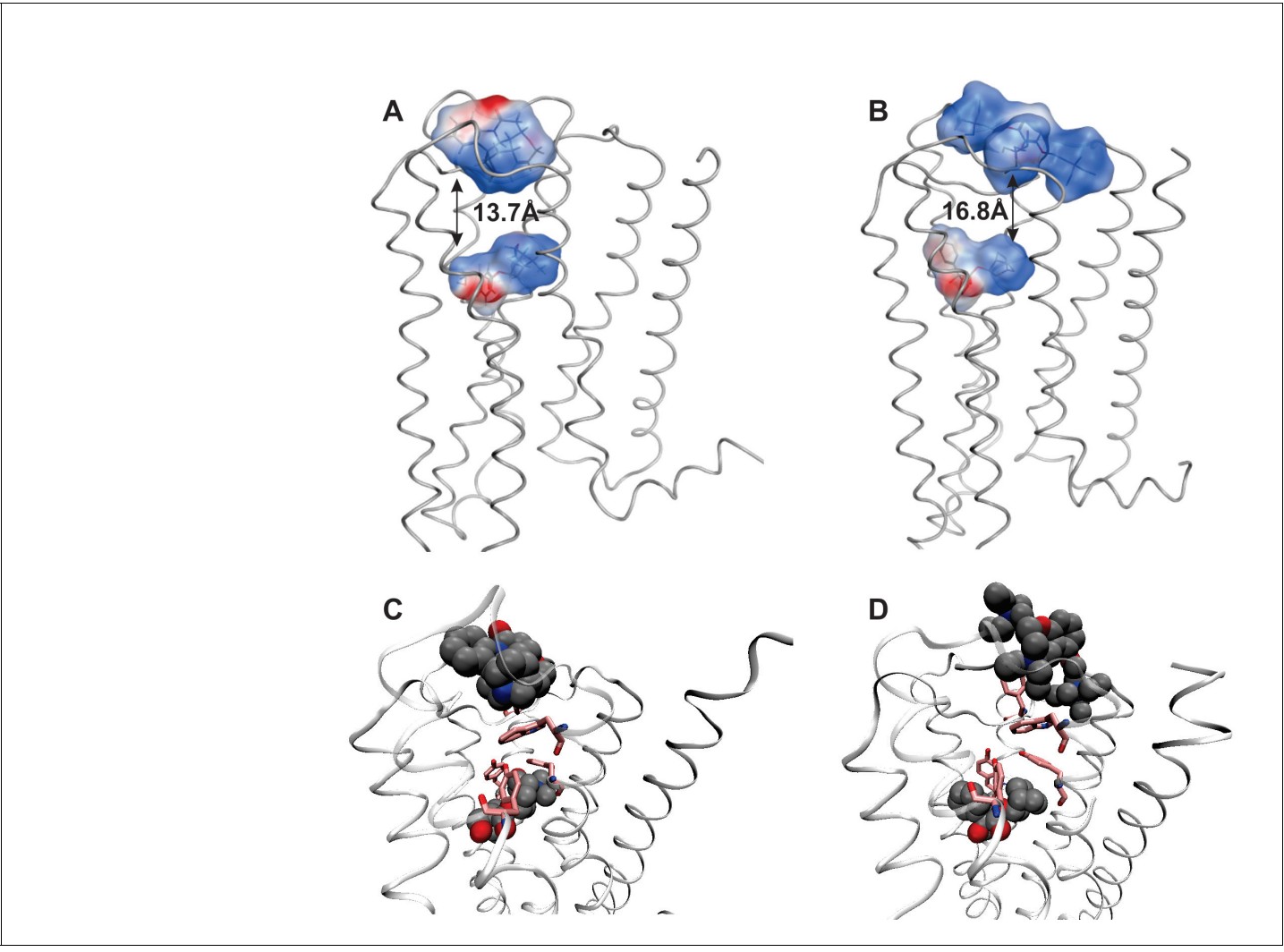

**Figure 4.** Intramolecular interactions between orthosteric and allosteric ligands. Models of the liganded receptor were simulated by molecular dynamics as described in Materials and Methods. The region of the ligand-binding sites is shown in the figure, with NMS and strychnine (**A**, **C**) or with QNB and gallamine (**B**, **D**) at the ortho- and allosteric sites, respectively. (**A**, **B**) The electrostatic potential of each ligand is displayed on a molecular surface within 4.5 Å of the constituent atoms (positive potential, blue; negative potential, red). The values and corresponding arrows are the distances between the centers of the cationic ammonium groups of the orthosteric and allosteric ligands. The closest such group is shown in the case of gallamine. (**C**, **D**) The bound ligands are shown together with residues involved in receptor-ligand interactions.

The following figure supplement is available for figure 4:

**Figure supplement 1.** Chemical structures of allosteric and orthosteric ligands to the M$_2$ muscarinic receptor.

has three cationic groups and therefore experiences greater electrostatic repulsion overall, but each group is surrounded by sterically bulky and electron-rich ethyl groups that diminish the repulsive effect (*Figure 5—figure supplement 1*). Also, the smaller steric load born by the single cationic nitrogen atom of strychnine allows for positioning closer to the orthosteric site.

A network of interactions serves as a conformational link between the allosteric and orthosteric sites. Three tyrosine residues form a hydrogen-bonded aromatic cap over the orthosteric site (*i.e.*, Tyr[104], Tyr[403], and Tyr[426]) (*Kruse et al., 2013*). Between the cap and the allosteric site is a tryptophan residue that interacts with the cationic nitrogen atom of the allosteric ligand (*i.e.*, Trp[422]). Tyrosine 426 within the aromatic cap and Trp[422] are linked via the peptide backbone of helix 7 (*Figure 5*). In the absence of an orthosteric ligand, the orientation of Trp[422] is unfavorable for the interaction with a cationic allosteric ligand (*Figure 5A*).

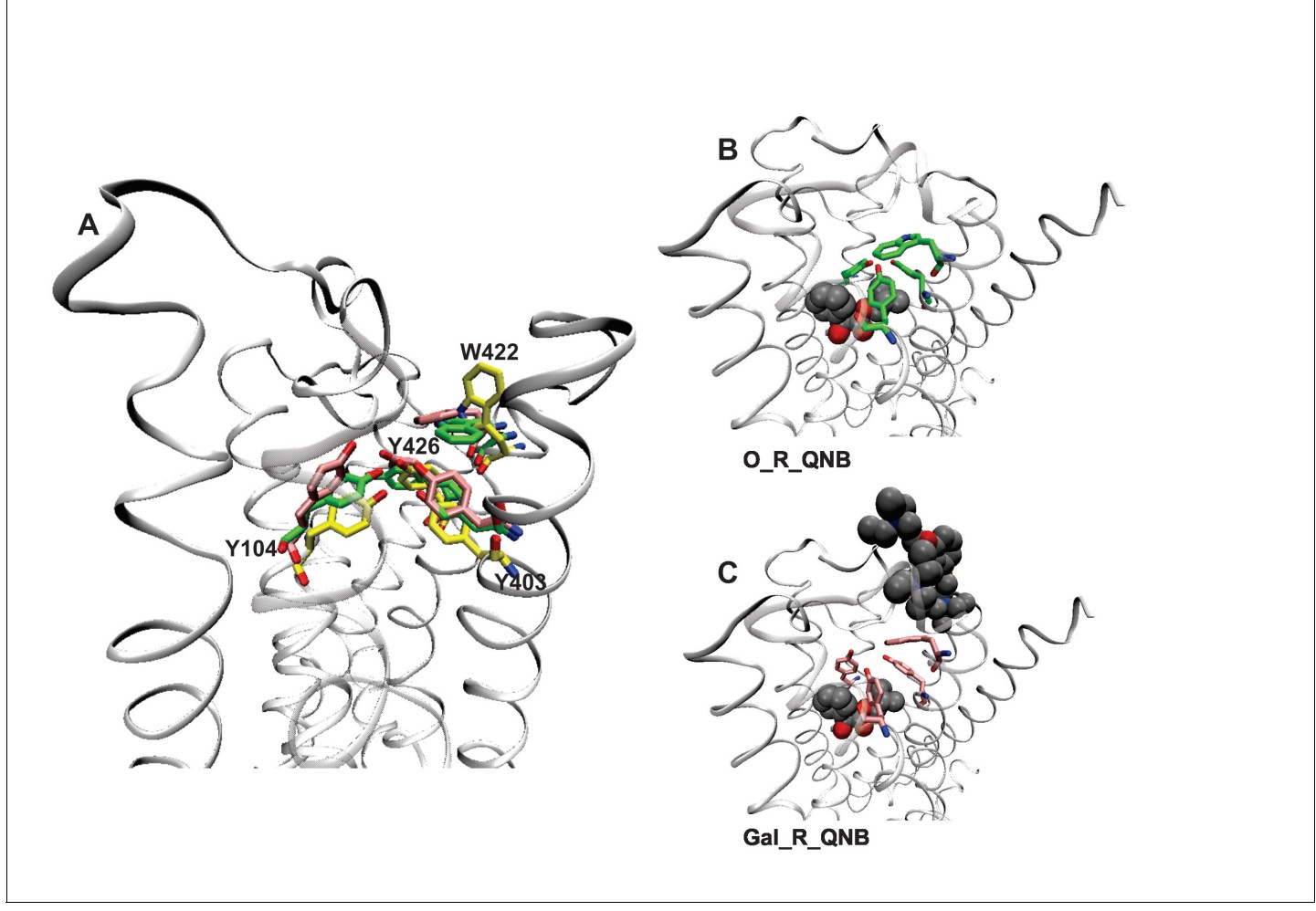

**Figure 5.** Effects of orthosteric and allsteric ligands on the aromatic cap and Trp[422]. (**A**) The three residues of the aromatic cap (*i.e.*, Tyr[104], Tyr[403], and Tyr[426]) and Trp[422] are shown in an overlay of the results of three simulations: a vacant receptor (yellow), a QNB-bound receptor (green), and a receptor occupied by both QNB and gallamine (pink). The ligands are not shown. (**B**) The receptor with QNB at the orthosteric site. (**C**) The receptor with QNB at the orthosteric site and gallamine at the allosteric site. Here and elsewhere, different liganded states of the receptor are identified as X_R_Y, where X and Y are the allosteric and orthosteric ligands, respectively. Occupancy is represented as O for a vacant site or as Str (strychnine), Gal (gallamine), NMS, or QNB for a ligand-occupied site.

The following source data and figure supplements are available for figure 5:

**Source data 1.** Figure 5–figure supplement 2–Distance between the $\alpha$-carbon atoms of Tyr[177] and Asn[419] in crystal structures of the $M_2$ receptor

**Source data 2.** Figure 5–figure supplement 3–Mean distances (Å) between the $\alpha$-carbon atoms of Tyr[177] and Asn[419] in the $M_2$ receptor with different combinations of allosteric and orthosteric ligands

**Figure supplement 1.** Orientation of ligands at the orthosteric site.

**Figure supplement 2.** Width of the vestibule to the orthosteric site.

**Figure supplement 3.** Distribution of distances between the $\alpha$-carbon atoms of Tyr[177] and Asn[419].

Binding of NMS or QNB at the orthosteric site reduces the mobility of the residues of the aromatic cap (*e.g.*, *Figure 5A and B*) and orients Trp[422] to interact with the cationic group of an allosteric ligand. The effect is greater with QNB than with NMS owing to the additional phenyl ring of QNB. Binding of an allosteric ligand draws the aromatic cap outward, leading to destabilization of

the orthosteric ligand and greater flexibility in the region of the orthosteric site (*e.g.*, *Figure 5A and C*). The effect is greater with gallamine than with strychnine, owing to the greater size and charge of gallamine, and it is suggestive of a reduction in the affinity of the receptor for orthosteric ligands.

Gallamine and strychnine affect the conformation of the receptor through interactions between the cationic ammonium group of each allosteric ligand and the side-chains of $Trp^{422}$ and $Tyr^{177}$. Gallamine also affects the receptor through $\pi$-cation interactions with $Tyr^{80}$ and $Tyr^{83}$ and through electrostatic interactions with $Glu^{172}$ of the EDGE motif. The overall conformational effect of these interactions was tracked by measuring the distance between the $\alpha$-carbon atoms of $Tyr^{177}$ and $Asn^{419}$. That distance is a measure of the width of the vestibule (*e.g.*, *Figure 5—figure supplement 2*), and it differs among crystal structures of the $M_2$ receptor in different liganded states (*Figure 5—source data 1*): namely, with QNB in the orthosteric site (3UON) (*Haga et al., 2012*), with iperoxo in the orthosteric site (4MQS), and with iperoxo in the orthosteric site and LY2119620 in the allosteric site (4MQT) (*Kruse et al., 2013*).

The width of the vestibule in simulations with NMS or QNB at the orthosteric site was distributed as shown in *Figure 5—figure supplement 3*. The allosteric site was vacant (*e.g.*, O_R_NMS) or occupied by strychnine or gallamine (*e.g.*, Str_R_NMS or Gal_R_NMS). The mean distances obtained for all nine liganded and unliganded states are listed in *Figure 5—source data 2*. When the allosteric site is vacant or occupied by strychnine, the vestibule is wider with QNB than with NMS at the orthosteric site. The greater width is due to the additional phenyl ring of QNB, which disrupts the positions of $Tyr^{403}$, $Trp^{422}$, and $Asn^{419}$. The vestibule is narrowest with strychnine and NMS. Other combinations of ligands have little or no effect owing to the bulk of QNB, the charge on gallamine, or both.

Destabilisation of the aromatic cap plus the electrostatic repulsion that exists with all positively charged allosteric–orthosteric pairs seems likely to prevent the simultaneous binding of both ligands to a monomeric receptor. The least disruptive effects of one ligand on the binding of another occurred with NMS and strychnine owing, in part, to conformational flexibility afforded by one ligand at the site of the other. The most disruptive effects occurred with QNB and gallamine owing to interactions involving the second phenyl ring of the former and the additional cationic centers of the latter (*Figure 4C and D*).

## Discussion

Allostery at the $M_2$ muscarinic receptor generally has been understood in terms of interactions within a monomer (*Christopoulos et al., 2002*), but that view disregards the tendency of GPCRs to form oligomers (*Park et al., 2004*). Moreover, the oligomeric nature of the $M_2$ receptor is evident in the multiphasic effects of allosteric ligands on the dissociation rate of orthosteric antagonists and their binding at equilibrium (*Shivnaraine et al., 2012*) (*e.g.*, *Figure 1*). Such effects appear to involve at least four allosteric sites in a mix of intra- and intermolecular heterotropic interactions within a complex of receptors that is tetrameric or larger (*Shivnaraine et al., 2012*).

We show here that all such complexity is lost when the $M_2$ receptor is purified as a monomer (*Figure 1*). Gallamine and strychnine were strictly inhibitory in their effect on the binding of [³H]NMS and [³H]QNB, and the rate of dissociation of [³H]QNB was decreased at all concentrations of gallamine. In each case, the dose-dependence was monophasic with a Hill coefficient of 1. Ligands therefore bind to monomers in the manner expected for a protein with one allosteric site and one orthosteric site, in contrast to the behavior of $M_2$ receptors in myocardial membranes and unprocessed solubilized preparations.

With either gallamine or strychnine, the potency defined by the single phase observed with monomers is similar to that defined by the weakest of the two or three components of the multiphasic curves observed with oligomers. It follows that effects associated with allosteric sites of higher affinity are a property of oligomers and derive from intermolecular interactions among the constituent protomers. They include the positive effect of gallamine on the rate of dissociation of [³H]QNB (*Figure 1C*) and varied effects on the binding of [³H]NMS at equilibrium: namely, negative cooperativity at sites of high affinity for gallamine (*Figure 1D*; log $K_1$ = −5.69, *Figure 1—source data 1*), apparent positive cooperativity at sites of intermediate affinity for gallamine (*Figure 1D*; log $K_2$ = −4.59, *Figure 1—source data 1*), and positive cooperativity at sites of high affinity for strychnine (*Figure 1E–G*).

In the case of gallamine, two further lines of evidence from CHO cells confirm the existence of heterotropic interactions between neighboring protomers and the relationship between those interactions and the allosteric sites of higher affinity. First, mutants lacking either the allosteric site or the orthosteric site were copurified from cotransfected cells to obtain a heteromer that lacks the capacity for intramolecular interactions. The purified complex retained the sites of high and intermediate affinity for gallamine that are seen in native preparations of the wild-type receptor but not the site of low affinity (*Figure 3E,F*); whereas the former are associated with oligomers, only the latter is observed in monomers. Second, the orthosteric site was eliminated in an $M_2$ receptor bearing mCherry at the *N*-terminus and FlAsH in ECL2, which together serve as a FRET-based sensor of conformation at the allosteric site (*Figures 2A*). The FRET efficiency of the binding-deficient mutant was affected by NMS only when the mutant was co-expressed with the wild-type receptor (*Figure 3D*).

Three lines of evidence indicate that the FRET-based sensor reports primarily on changes at the allosteric site. A comparison of the fluorescence anisotropies measured for eGFP fused to the *N*-terminus of the $M_2$ receptor suggests that the region of the fusion is comparatively rigid and unaffected by orthosteric ligands. In contrast, the FRET efficiency of the sensor expressed alone in CHO cells was increased or decreased by orthosteric ligands in a manner that tracked the pharmacological identity of the ligand as an agonist, a partial agonist, or an inverse agonist (*Figure 2F*). Ligand-dependent changes in FRET between FlAsH and mCherry therefore appear to result from changes in the position of ECL2-bound FlAsH. Finally, NMS increased the FRET efficiency by 20 percentage points. Such a change is consistent with the predictions of molecular dynamics simulations for the effect of NMS on the width of the vestibule to the orthosteric site (*Figure 5—source data 1*) (*Dror et al., 2013*).

Allosteric and orthosteric ligands may interact via steric hindrance, conformational changes, or electrostatic effects. In the structure computed for a monomeric $M_2$ receptor, the interaction between gallamine or strychnine on the one hand and NMS or QNB on the other is dominated by electrostatic repulsion (*Figure 4*, *Table 1*). That explains why gallamine and strychnine were inhibitory in their effect on the binding of [$^3$H]NMS to purified monomers (*Figure 1B and C*), and it suggests a molecular basis for the weak affinities of both ligands in those assays. It also suggests that electrostatic repulsion dominates intramolecular cooperativity between allosteric and orthosteric ligands within the constituent protomers of an oligomer.

The molecular dynamics simulations place each allosteric ligand at the extracellular surface, where it caps the vestibule to the orthosteric site and sterically hinders passage of the orthosteric antagonist (*Figure 4*). This arrangement is consistent with crystallographic data on the location of a positive allosteric modulator, LY2119620 (*Kruse et al., 2013*), and with the kinetics of allosteric modulation. A kinetically defined model that describes capping in a monomeric receptor (*Figure 6*) predicts that the rate at which the system equilibrates will depend upon the order in which the ligands are added to the receptor (*Figure 1E–G*). The predicted effects were observed experimentally with purified monomers of the $M_2$ receptor, in that equilibration was slowest when [$^3$H]NMS preceded strychnine (*Figure 1—figure supplement 2*).

Allosteric effects in monomers are monophasic, necessarily intramolecular, exclusively negative, and of comparatively low affinity. Those in oligomers may be positive or negative and generally reveal two or more affinities, the weakest of which corresponds to that in monomers. The versatility of oligomers is a consequence of additional mechanistic pathways made possible by intermolecular interactions, which avoid the electrostatic repulsion that dominates intramolecular interactions and enforces negative cooperativity within monomers. Such intermolecular effects presumably are mediated by ligand-sensitive conformational changes transmitted from one protomer of receptor to another.

Although the details of allosteric modulation cannot be determined through simulations based solely on a monomer, some insight into the effects observed with different ligand-pairs may be gained by monitoring the residues that form an aromatic cap over the orthosteric site (*i.e.*, Tyrosines 104, 403, and 426). Positive allosteric modulation by strychnine (*Figure 1E*), negative modulation by gallamine at sites of high affinity, and apparent positive modulation by gallamine at sites of intermediate affinity (*Figure 1D*) occurred only with NMS as the radioligand. It appears, however, that the positive modulatory effects of strychnine and gallamine are different in kind. Whereas strychnine caused a net increase in the binding of [$^3$H]NMS, gallamine never raised the level of binding above that in its absence.

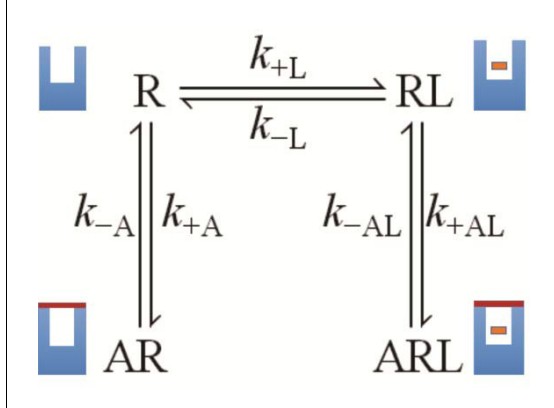

**Figure 6.** A receptor (R) binds an orthosteric ligand (L) and an allosteric ligand (A) to form a ternary complex (ARL). Each ligand can bind separately to form AR or RL, but the ternary complex is accessible only via RL. The orthosteric site of the $M_2$ receptor is located within the cluster of helical domains, with access via a vestibule that forms the allosteric site. Occupancy of the latter by an allosteric ligand precludes association and dissociation of the orthosteric ligand. The parameters $k_{-L}$ and $k_{+L}$ are the first- and second-order rate constants for the binding of L to R; similarly, $k_{-A}$ and $k_{+A}$ are the first- and second-order rate constants for the binding of A to R. The parameters $k_{-AL}$ and $k_{+AL}$ are the rate constants for the binding of A to RL.

The effect of strychnine is a clear example of positive cooperativity, but that of gallamine appears to be an attenuation of negative cooperativity. Strychnine and NMS therefore were the only ligand-pair to display positive cooperativity that was self-evident and unambiguous. They also are the ligand-pair that caused minimal disruption of the aromatic cap in molecular dynamics simulations, and positive cooperativity may be a consequence of attendant conformational flexibility in that region of the receptor. The least flexibility was observed with QNB and gallamine, which engaged only in negative cooperativity.

A similar pattern is seen in the distance between the $\alpha$-carbon atoms of Tyr[177] and Asn[419], which is a measure of the width of the vestibule that accommodates the allosteric ligand. Decreases in the width from that in the vacant receptor (11.4 Å) were observed only with NMS at the orthosteric site of an otherwise vacant receptor (10.9 Å) and only with strychnine at the allosteric site of a NMS-liganded receptor (10.1 Å). There was little effect of QNB alone or of other ligand-pairs.

Direct evidence for interactions between the protomers of an oligomer has been obtained by precluding intramolecular effects in studies involving the FRET-based sensor and the binding of [³H]NMS, and molecular dynamics simulations have provided some insight into the underlying conformational effects. Taken together, the results suggest that intermolecular interactions are associated with narrowing of the vestibule and conformational flexibility around the aromatic cap. Such an arrangement would allow an allosteric ligand to bind with high affinity to protomers with a vacant orthosteric site and thereby to increase the binding of an orthosteric ligand to linked protomers with a vacant allosteric site.

The role of oligomers formed by GPCRs of Family 1A is debated, in part because oligomers are not seen to possess a unique or obligatory functionality. Both monomers and oligomers can activate G proteins, and both display allosteric communication between agonists at the receptor and guanylyl nucleotides at the G protein (*Redka et al., 2014*). In the case of the $M_2$ muscarinic receptor, however, the two forms mediate those effects by different mechanisms. Only oligomers mimic the binding patterns observed in native membranes (*Redka et al., 2014*). We show here that oligomers of the $M_2$ receptor also are responsible for the high-affinity binding of allosteric ligands and for positive allosteric modulation. Whereas both effects are observed routinely with $M_2$ receptors in natural membranes, neither is expected or observed with monomers. These observations illustrate the cooperative capability of oligomers formed by GPCRs, and they add to the complexity that must be accommodated in a mechanistic understanding of events at the level of the receptor.

## Materials and methods

### Construction of plasmids

Substitutions, insertions, and deletions of bases were performed by site-directed mutagenesis (Quick-Change, Agilent Technologies). PAGE-purified primers were obtained from Integrated DNA Technologies (IDT). Constructs were prepared in pcDNA3.1 for expression in CHO cells and in Bac-N-Blue (Life Technologies) for expression in *Sf9* cells. The human $M_2$ muscarinic receptor was used

throughout, and all sequences were confirmed by DNA sequencing (Centre for Applied Genomics, Hospital for Sick Children, Toronto).

## Fluorophore-tagged variants of the M$_2$ receptor

The receptor was fused at the N-terminus to eGFP (eGFP-M$_2$) or mCherry (mCh-M$_2$). To obtain a FRET-based sensor with fluorescein arsenical hairpin binder (FlAsH) as the donor, the second extracellular loop (ECL2) of mCh-M$_2$ was modified by insertion of the FlAsH-reactive sequence MLMCCPGCCMEP (FCM) between Val$^{166}$ and Gly$^{167}$(mCh-M$_2$-FCM). Similarly, mCherry was inserted between Val$^{166}$ and Gly$^{167}$ in eGFP-M$_2$. To assess the rotational motion of a fluorophore at the N-terminus, eGFP was fused to the full-length receptor (eGFP-M$_2$) and to a truncated mutant lacking the first 13 amino acids (eGFP-truncM$_2$) (forward primer, 5'-GGCATGGACGAGCTGTACAAGAATAG TCCTTATAAGAC-3'). Further details are provided in Appendix 1

## Binding-defective mutants of the M$_2$ receptor

To preclude binding to the allosteric site, negatively charged residues in the EDGE sequence at positions 172–175 of ECL2 were replaced by positively charged residues (i.e., Lys$^{172}$Arg$^{173}$Gly$^{174}$Lys$^{175}$). To preclude binding to the orthosteric site, aspartic acid at position 103 was replaced by alanine in the wild-time receptor [M$_2$(D103A)] and the sensor [mCh-M$_2$(D103A)-FCM]. Further details are provided in Appendix 1.

## **Expression, extraction, and purification of the M$_2$ receptor**

### Sf9 cells

Human M$_2$ muscarinic receptor bearing the c-Myc or FLAG epitope at the N-terminus was expressed in Sf9 cells as described previously (*Redka et al., 2014*; *Shivnaraine et al., 2012*) and in Appendix 1. The cells were harvested and solubilized in digitonin–cholate (0.86% digitonin, Wako Chemicals USA; 0.17% cholate, Sigma-Aldrich), and aliquots of the extract were removed for electrophoresis and binding assays. The receptor was purified in predominantly monomeric form by successive passage on DEAE-Sepharose, ABT-Sepharose, and hydroxyapatite. The final concentrations of digitonin and cholate were 0.1% and 0.02%, respectively. Purified receptor was stored at −75°C. Further details regarding the purification and the nature of the purified receptor have been described previously (*Redka et al., 2014*).

### CHO cells

Chinese Hamster Ovary (CHO) cells stably expressing the wild-type human M$_2$ receptor were grown and processed as described previously (*Redka et al., 2014*; *Shivnaraine et al., 2012*). Cells for transient transfections were grown and processed as described in Appendix 1.

To obtain membranes for binding assays, thawed cells containing wild-type M$_2$ receptor or mCh-M$_2$-FCM were homogenized and processed as described previously (*Redka et al., 2014*; *Shivnaraine et al., 2012*). To obtain extracts for measurements of binding or fluorescence anisotropy, thawed cells containing mCh-M$_2$-FCM, M$_2$-ECL2+ve, M$_2$-ECL2+ve plus M$_2$(D103A), eGFP-M$_2$, or eGFP-truncM$_2$ were washed in buffer A [20 mM HEPES, 20 mM NaCl, 1 mM EDTA, 0.1 mM PMSF, Complete Protease Inhibitor Cocktail tablets (Roche, 1 tablet/50 mL), adjusted to pH 7.40 with NaOH] and centrifuged for 10 min at 3,000 × g and 4°C. The washed cells were resuspended in buffer A, and the mixture was homogenized with three bursts of a Brinkman Polytron (setting 6, 10 s). An aliquot of 100 μL was removed for the determination of total protein. The homogenate then was centrifuged for 30 min at 45,000 × g and 4°C, and the pellet was resuspended in buffer A (5.5 g of protein per L) with three bursts of the Polytron (setting 6, 10 s). Solubilization was initiated by the addition of digitonin and sodium cholate to final concentrations of 0.86% and 0.17%, respectively, and the mixture was agitated on a rocking platform for 15 min at room temperature. The sample then was diluted 1:1 in buffer A and centrifuged for 45 min at 45,000 × g and 4°C. The supernatant fraction was concentrated (Amicon Ultra-4, 30 kDa, Millipore) and divided into aliquots that were stored at −75°C (*Redka et al., 2014*; *Shivnaraine et al., 2012*). Such extracts from cells expressing mCh-M$_2$-FCM or M$_2$-ECL2+ve were characterized for the binding of [$^3$H]NMS at graded concentrations of the radioligand.

To obtain purified oligomers containing both FLAG-tagged $M_2$(D103A) and $His_6$-tagged $M_2$-ECL2 +ve, the solubilized receptor was applied successively to columns of $Ni^{2+}$-nitriloacetic acid (NTA)-agarose (Qiagen) and anti-FLAG Sepharose (Sigma-Aldrich). The columns were pre-equilibrated with buffer A supplemented with digitonin (0.1%) and cholate (0.04%), and the receptor was eluted with the same buffer containing imidazole (150 mM) (Sigma-Aldrich) or FLAG peptide (100 µg/mL) (Sigma-Aldrich), respectively.

## Porcine atria

The $M_2$ receptor is the predominant muscarinic subtype in porcine atria (*Wreggett et al., 1995*). It was extracted from sarcolemmal membranes according to a two-step procedure in which the membranes were resuspended in buffer B (20 mM imidazole, 1 mM EDTA, 0.1 mM PMSF, 0.02% $NaN_3$, adjusted to pH 7.60 with HCl) supplemented with digitonin (0.36%) and cholate (0.08%) (5.5 g of total protein per L), recovered by centrifugation, and resuspended in buffer B supplemented with digitonin (0.8%) and cholate (0.08%). The soluble fraction from the second resuspension, which contained the $M_2$ receptor, was stored at −70°C until required for binding assays. Further details have been described previously (*Shivnaraine et al., 2012*) and are summarized in Appendix 1.

## Cross-linking, immunoprecipitation, and electrophoresis

### Cross-linking

A solution of the cross-linker *bis*(sulfosuccinimidyl)suberate ($BS^3$, Pierce) in deionized water (20 mM) was added to an aliquot of the receptor to yield a final reagent concentration of 2 mM. The mixture was incubated for 30 min at 24°C, and the reaction was terminated by the addition of Tris-HCl (1 M, pH 8.00) to a final concentration of 20 mM. After further incubation for 15 min at 24°C, the sample was stored on ice prior to electrophoresis. Controls lacking $BS^3$ were prepared in parallel under otherwise identical conditions.

### Co-immunoprecipitation

An aliquot of the extract (500 µL) from *Sf9* cells co-expressing the FLAG- and c-Myc-tagged receptors was supplemented with a 50% slurry (20 µL) of agarose-conjugated anti-FLAG anti-body (Santa Cruz Biotechnology, Inc.). The mixture was shaken overnight at 4°C, and immunoadsorbed receptor was collected by centrifugation. The precipitated beads then were washed 4 times with 3 mL of buffer C (20 mM HEPES, 1 mM EDTA, 0.1 mM PMSF, adjusted to pH 7.40 with NaOH) supplemented with digitonin (1%) and cholate (0.001%), and the entire precipitate was applied to the polyacrylamide gel. Following electrophoresis and transfer, the nitrocellulose membrane (Bio-Rad, 0.45 µm) was blotted with anti-c-Myc antibody (Santa Cruz Biotechnology, Inc.) as described previously (*Ma et al., 2007*).

### Electrophoresis and western blotting

Details regarding these procedures are described Appendix 1.

## Labeling of the $M_2$ receptor with FlAsH in live cells

TC-FlAsH was obtained as a kit from Molecular Probes (Invitrogen), and labeling was carried out under reduced light according to a procedure adapted from the manufacturer's instructions (*Hoffmann et al., 2010*). Transfected CHO cells growing at 50–75% confluency were washed twice with PBS, and the medium was changed to Opti-MEM reduced serum medium (Life Technologies, Inc.). After further incubation of the cells for 1 hr at 37°C, the medium was removed and replaced by a freshly prepared labeling solution of FlAsH in reduced serum medium (2.5 µM, 1 mL per dish). The culture was incubated for 20 min at 37°C in 5% $CO_2$; the labeling solution then was removed, and the cells were washed twice in a buffer containing 500 µM BAL (Life Technologies, Inc.). During the second wash, the BAL buffer was left for 5 min at 37°C prior to its removal. Washed cells were prepared for imaging by the addition of DMEM (2 mL) supplemented with 50 mM HEPES at pH 7.40 (Gibco Life Technologies, Inc.).

## Binding assays

$N$-[³H]Methylscopolamine ([³H]NMS, 87 Ci/mmol) and (−)-[³H]quinuclidinylbenzilate ([³H]QNB, 42 Ci/mmol) were purchased from PerkinElmer as a solution in ethanol, which was removed by evaporation prior to use. Strychnine, gallamine, carbachol, pilocarpine, and unlabeled NMS and QNB were purchased from Sigma-Aldrich. All other reagents were from the sources described previously (*Shivnaraine et al., 2012*).

Binding was measured at pH 7.40 in buffer C or buffer D (Dulbecco's phosphate-buffered saline, Sigma-Aldrich D5652, supplemented with 1 mM $CaCl_2$ and 1 mM $MgCl_2$). In the case of detergent-solubilized preparations, buffer C was supplemented with digitonin (1%) and cholate (0.02%). The separation of free and bound radioligand was achieved by chromatography on Sephadex G-50 Fine in the case of detergent-solubilized receptor and by filtration on fiberglass filters or microcentrifugation in the case of membrane-bound receptor. Further details have been described previously (*Shivnaraine et al., 2012*).

The net dissociation of [³H]QNB over time was analyzed in terms of a single exponential according to *Equation 1*, in which $k_{obsd}$ is the rate constant; $B_{obsd}$ represents total binding of the radioligand at time t, and $B_{t=0}$ and $B_{t→∞}$ are the initial and asymptotic levels of binding, respectively.

$$B_{obsd} = (B_{t=0} - B_{t→∞})e^{-k_{obsd}t} + B_{t→∞} \qquad (1)$$

Time-courses at one or more concentrations of gallamine were accompanied in the same experiment by a control lacking the allosteric ligand. The data from all traces were analyzed in concert with a single value of $B_{t→∞}$ and separate values of $k_{obsd}$ and $B_{t=0}$. The constraint on $B_{t→∞}$ was without appreciable effect on the sum of squares (p>0.05). The value of $k_{obsd}$ measured in the absence of gallamine was designated $k_0$ and used to normalize each value measured in the presence of gallamine (*i.e.*, $k_{obsd}/k_0$).

Dose-dependent effects of an allosteric modulator (A) on the normalized rate of dissociation of the radioligand ($k_{obsd}/k_0$) or on the level of total binding at a specified time ($B_{obsd}$) were analyzed empirically in terms of *Equation 2*.

$$Y_{obsd} = Y_{[A]→∞} + \left(Y_{[A]=0} - Y_{[A]→∞}\right) \sum_{j=1}^{n} \frac{F_j K_j^{n_{H(j)}}}{[A]_t^{n_{H(j)}} K_j^{n_{H(j)}}} \qquad (2)$$

$$\sum_{j=1}^{n} F_j = 1$$

Estimates of binding at graded concentrations of [³H]NMS were analyzed in terms of *Equation 3*.

$$B_{obsd} = B_{max} \frac{\left([P]_t - B_{sp}\right)^{n_H}}{K^{n_H} + \left([P]_t - B_{sp}\right)^{n_H}} + NS\left([P]_t - B_{sp}\right) \qquad (3)$$

The parameter $B_{max}$ represents the maximal specific binding of the radioligand (P), and $B_{obsd}$ and $B_{sp}$ represent total and specific binding, respectively, at the total concentration $[P]_t$; $n_H$ is the Hill coefficient, and $K$ is the concentration of unbound [³H]NMS that corresponds to half-maximal binding. NS is the fraction of unbound radioligand that appears as nonspecific binding. Fitted estimates of the Hill coefficient ranged from 0.93 to 1.03 and were indistinguishable from 1 (p>0.05).

Analyses in terms of *Equations 1–3* typically were performed on data from replicate experiments, which were taken in concert to obtain single fitted values of parameters that are expected to be invariant (*i.e.*, $k_{obsd}$, $K$, $n_H$). In figures that show the results of such analyses, data from individual experiments have been presented with reference to a single fitted curve. To obtain the values plotted on the y-axis, measured estimates of $B_{obsd}$ or $Y_{obsd}$ were adjusted according to the equation $Y' = Y[f(x_i, \bar{a}, b)/f(x_i, a, b)]$ (*Park et al., 2002*). The function $f$ represents the fitted equation, and $x_i$ represents the independent variable at point i. The vectors **a** and **b** represent fitted parameters that were estimated separately for each experiment (**a**) or as a single value common to all experiments (**b**); $\bar{a}$ is the corresponding vector in which parametric values that differed among different experiments have been replaced by the means. Individual values of $Y'$ at the same $x_i$ were averaged to obtain the mean and standard error plotted in the figure. Details regarding the optimization of parameters and statistical procedures are described in Appendix 1.

## Fluorescence, microscopy, and image-analysis

The rotational flexibility of eGFP fused to the $N$-termini of the wild-type $M_2$ receptor (eGFP-$M_2$) and a truncated mutant (eGFP-truncM$_2$) was measured by time-resolved fluorescence anisotropy of single molecules in a locally constructed instrument. Each fusion protein was expressed in CHO cells and solubilized at a concentration of 3–10 nM in buffer C supplemented with detergent (0.8% digitonin, 0.04% sodium cholate). Aliquots of the extract were applied to a coverslip and excited at 480 nm with an excitation beam obtained by frequency-doubling the output of a femtosecond laser (Tsunami HP, Spectra Physics, Santa Clara, CA, USA).

The emission from a confocal volume 5 μm above the surface was passed through a Plan-Apochromat oil-immersion objective (100×, Carl Zeiss, Canada) and projected onto an avalanche photodiode (APD). The emission was divided by means of a polarization cube into two beams with orthogonal polarizations and collected on two different detectors (PDM-5CTC, MPD, Milano, Italy). Photon arrival times were recorded at 4 ps resolution using a multichannel counter (PicoHarp300, PicoQuant, Germany).

The fluorescence signals from the two detectors were fit globally in MATLAB using custom-written software and a Levenberg-Marquardt algorithm with iterative re-convolution. Repetitive excitation and the color off-set ($s$) between measurements of the instrument response function (IRF) and the fluorescence were accounted for according to *Equations 4 and 5*, in which $d_{par}$ and $d_{perp}$ represent the parallel and perpendicular decays before convolution.

$$D_{par} = IRF_{par}, s \otimes d_{par} \tag{4}$$

$$D_{perp} = IRF_{perp}, s \otimes d_{perp} \tag{5}$$

The fluorescence signal in each plane was summed to obtain the total fluorescence at time t ($F$(t)), and the lifetime was obtained according to *Equation 6*.

$$F(t) = \frac{A}{1 - exp\left(-\frac{T}{\tau}\right)} exp\left(-\frac{t}{\tau}\right) \tag{6}$$

The parameter $A$ in *Equation 6* is the amplitude, and $T$ is the repetition time of the laser (*i.e.*, 12.5 ns); t is the time bin, and τ is the decay constant. Corrections for collecting the fluorescence emission in the parallel and perpendicular planes through a high numerical-aperture lens were performed according to *Equations 7 and 8*.

$$d_{perp} = \left(\frac{1}{3}\right)F(t)(1 + (1 - 3k_2)r(t)) \tag{7}$$

$$d_{par} = G\left(\frac{1}{3}\right)F(t)(1 + (2 - 3k_1)r(t)) \tag{8}$$

The correction factors $k_1$ and $k_2$ have values of 0.33 and 0.065, respectively. The correction factor G corrects for the difference in the sensitivity of detection between the two channels and was taken as 1.061, as determined from a solution of rhodamine 110 (10 nM). The rotational correlation time of eGFP ($\varphi$) was estimated from the loss of anisotropy ($r$) over time (t) according to *Equation 9*, in which $r$(0) is the amplitude of the decay.

$$r(t) = r(0)\left(exp\left(-\frac{t}{\varphi}\right)\right) \tag{9}$$

Where r0 is the is the decay amplitude and $\phi$ is the rotational correlation time.

Confocal imaging of intact CHO cells was performed on a Zeiss microscope (model LSM710) using a Plan-Apochromat oil-immersion objective lens (63×, 1.4 NA). Samples were irradiated at 488 nm and a power of 0.37 μW. An area of 134.7 × 134.7 μm$^2$ was captured through a pinhole of 1 Airy unit, and each pixel in the image represented 0.26 μm$^2$ according to the Nyquist theorem. A stack of 30 images was acquired at 5 nm intervals from 495 nm to 640 nm, and the data were processed

through custom-written software in MATLAB to obtain the emission spectrum corrected for background.

Individual spectra were unmixed by linear regression according to *Equation 10* (two colors) or 11 (three colors), in which $k_D$, $k_A$, and $k_B$, are the scaling factors for the contributions from FlAsH, mCherry, and eGFP, respectively.

$$Y = k_D Em_D + k_A Em_A \qquad (10)$$

$$Y = k_B Em_B + k_D Em_D + k_A Em_A \qquad (11)$$

The constants $Em_D$ and $Em_A$ represent the reference spectra for the donor (FlAsH) and acceptor (mCherry), respectively; $Em_B$ represents the reference spectrum for eGFP, which was used as a marker for receptor with an intact orthosteric site when co-expressed with a binding-defective mutant.

The unmixed values of $k$ for the donor ($k_D$) and the acceptor ($k_A$) were used to calculate the apparent FRET efficiency ($E_{app}$) according to *Equation 12* (*Patowary et al., 2013*; *Raicu, 2007*). In the case of $k_A$, the fitted value from *Equation 10* or *11* was reduced by 6% to compensate for the direct excitation of mCherry at 498 nm.

$$E_{app} = \frac{1}{1 + \frac{Q_A}{Q_D}\frac{k_D}{k_A}\frac{W_D}{W_A}} \qquad (12)$$

The constants $Q_D$ and $Q_A$ represent the quantum yields, which were taken as 0.70 for the donor and 0.22 for the acceptor (*Subach et al., 2009*); $W_D$ and $W_A$ are the corresponding spectral integrals, and the ratio of the regions between 495 and 640 ($W_D/W_A$) was computed from the reference spectra to be 0.94.

## Kinetically determined mechanistic modeling

Time-dependent binding of an orthosteric ligand (L) to a monomeric receptor (R) was simulated according to *Figure 6*, in which access to and egress from the orthosteric site is precluded by an allosteric ligand (A). The system is defined by four ordinary differential equations (*Equation 13*), which were solved numerically to obtain the amount of bound orthosteric ligand at different times (*i. e.*, [RL] + [ARL]). It was assumed that neither ligand is depleted through binding to the receptor, and the free concentrations in *Equation 13* were taken as equal to the total concentrations. The simulations were performed in MATLAB 2012b, and the integrals were calculated using the ODE23s subroutine.

$$
\begin{aligned}
\frac{d[RL]}{dt} &= (k_{+L}[R][L] - k_{-L}[RL]) + (k_{-AL}[ARL] - k_{+AL}[RL][A])\\
\frac{d[AR]}{dt} &= (k_{+A}[R][A] - k_{-A}[AR])\\
\frac{d[ARL]}{dt} &= (k_{+AL}[RL][A]) - (k_{-AL}[ARL])\\
\frac{d[R]}{dt} &= (k_{-L}[RL] - k_{+L}[R][L]) + (k_{-A}[AR] - k_{+A}[R][A])
\end{aligned}
\qquad (13)
$$

The rate constants in *Equation 13* were computed from the affinity constants according to *Equations 14–16*, as described previously (*Shivnaraine et al., 2012*).

$$K_L = \frac{[R][L]}{[RL]} \equiv \frac{k_{-L}}{k_{+L}} \qquad (14)$$

$$K_A = \frac{[A][R]}{[AR]} \equiv \frac{k_{-A}}{k_{+A}} \qquad (15)$$

$$K_{AL} = \frac{[A][RL]}{[ARL]} \equiv \frac{k_{-AL}}{k_{+AL}} \equiv \alpha K_A \qquad (16)$$

Values of the cooperativity factor $\alpha$ were partitioned between the rate constants for association and dissociation according to *Equation 17*. The value of $j$ was 2 throughout.

$$\frac{k_{-\mathrm{AL}}}{k_{+\mathrm{AL}}} = \alpha \frac{k_{-\mathrm{A}}}{k_{+\mathrm{A}}} \equiv \frac{\alpha^{j+1} k_{-\mathrm{A}}}{\alpha^{j} k_{+\mathrm{A}}} \equiv \frac{\left(\frac{1}{\alpha}\right)^{j} k_{-\mathrm{A}}}{\left(\frac{1}{\alpha}\right)^{j+1} k_{+\mathrm{A}}} \tag{17}$$

## Molecular modeling and dynamics

Simulations were performed in the isothermal-isobaric (NPT) ensemble according to the Nosé-Poin-caré-Andersen (NPA) equations of motion with a time-step of 2 fs. A cut-off of 10 Å was applied to non-bonding interactions. Equilibration was for 1 ns (300 K, harmonic restraints of 0.5 kcal mol$^{-1}$ Å$^{-2}$ applied to non-hydrogen atoms), and production continued for 30 ns (300 K, no restraints). Water molecules were wrapped, bond-lengths to lone-pairs, and hydrogen atoms were constrained. Systems were sampled, atomic coordinates saved, and snapshots taken every 2.5 ps. Images were rendered using either MOE or VMD.

### Construction of the model for mCh-M$_2$-FCM

Crystal structures of the M$_2$ muscarinic receptor (4MQS, 4MQT) and mCherry (2H5Q) were obtained from the Protein Data Bank. The FlAsH-reactive sequence (FLNCCPGCCMEP) was incorporated into ECL2 of the receptor by homology modeling (Modeller 9.13). While modeling the loop, the rest of the protein was constrained to the crystallographic conformation. FlAsH was built in MOE 2013.08 (Molecular Operating Environment, 2014) and attached to the insertion sequence (CCPGCC) via covalent bonds between the sulfur atoms of the cysteine residues and the arsenic atoms of FlAsH. The length of the sulfur-to-arsenic bonds was set at 2.275 Å. Unresolved regions at the C-terminus of mCherry, the N-terminus of the receptor, and within the intracellular loop of the receptor between transmembrane domains 5 and 6 were modelled as described in Appendix 2

### Ligand–receptor and ligand–ligand interactions

The structure of the receptor and the structures of orthosteric and allosteric ligands were computed as described in Appendix 2 . NMS or QNB was inserted at the orthosteric site by induced-fit docking using MOE, and the initial placement was guided by alpha spheres within 5 Å of iperoxo in 4MQS. The conformers generated previously for each ligand were docked using the Triangle Matcher placement method and the London dG initial scoring function. A post-placement force-field refinement then was carried out, allowing the receptor and the ligand to move freely within a cut-off distance of 10 Å for receptor-ligand interactions. Final poses were evaluated according to the GBVI/WSA dG scoring function. Structures were verified by superimposition on the 3UON crystal structure of the M$_2$ receptor with QNB at the orthosteric site (Haga et al., 2012) guided by the essential interactions between the ligand and Asp[103], Tyr[104], Tyr[403], Asn[404], and Tyr[426], as displayed in 4MQS and 3UON, and by comparison with the results of previous modeling studies (Dror et al., 2013).

Strychnine or gallamine was inserted at the allosteric site in the manner described above for ligands at the orthosteric site. The site was defined by those amino acids in proximity with the allosteric ligand LY2119620 in the 4MQT structure. In the top-ranked poses, the cationic ammonium groups of strychnine and gallamine were oriented to interact with Tyr[177] and Trp[422]. The second ammonium group of gallamine lay in the vicinity of Tyr[80] and Tyr[83], with the third ammonium group extending into solvent. There was no pose in which cationic groups interacted with the EDGE sequence of ECL2, although this interaction appeared during molecular dynamics simulations. The poses of gallamine and strychnine agreed with those obtained in previous simulations (Dror et al., 2013).

The minimized structures of the vacant receptor and the eight complexes resulting from insertion of the four ligands were used as the starting points for molecular dynamics simulations in MOE. All atoms were represented explicitly using the CHARMM27 force field. Ligand-related parameters were checked using the CGenFF program on the CHARMM ParamChem server. Each system was solvated explicitly in a periodic box of TIP3P water molecules (approximately 12,800 molecules), neutralized with sodium chloride, and minimized to an RMS gradient of <1.0 prior to simulation in MOE. The resulting system had approximately 45,800 atoms and a density of 1.009 g/cm$^3$.

For each allosteric–orthosteric pair, the distance between their cationic nitrogen atoms was estimated according to Coulomb's law. In the case of gallamine, the nitrogen atom closest to that of the orthosteric ligand was selected. The electrostatic potential of an allosteric ligand in the absence

and presence of an orthosteric ligand was calculated using the CHARMM27 force field, and the difference was taken as a measure of the energetic cost of repulsion. A degree of electrostatic repulsion was observed with all four ligand-pairs. The effect was observed throughout the simulations, but it diminished over time as the cations compensated by moving apart. In the case of strychnine and QNB, for example, the average difference in electrostatic potential was 0.7 kcal/mol over the first 10 ns of production and 0.06 kcal/mol over a period of 30 ns. Because the average electrostatic potential of the ligands over time does not scale directly with the degree of repulsion, the result of repulsion was tracked in terms of distance.

## Acknowledgements

We are grateful to the managers and staff of Quality Meat Packers Ltd. for generous supplies of porcine atria. We thank Dr. Stephane Angers of the Leslie Dan Faculty of Pharmacy for helpful discussions over the course of the investigation and Dr. Donald F. Weaver of the Krembil Research Institute, University Health Network, for providing the facilities used for molecular dynamics simulations.

## Additional information

### Funding

| Funder | Grant reference number | Author |
| --- | --- | --- |
| Natural Sciences and Engineering Research Council of Canada | 371705 | Jonathan V Rocheleau |
| Natural Sciences and Engineering Research Council of Canada | 316271 | Claudiu C Gradinaru |
| National Institutes of Health | R01AG05214 | John Ellis |
| Pennsylvania Department of Health | Tobacco CURE funds 4100062216 | John Ellis |
| Canadian Institutes of Health Research | MOP97978 | James W Wells |
| Heart and Stroke Foundation of Canada | NA7168 | James W Wells |
| Heart and Stroke Foundation of Canada | G140006078 | James W Wells |

The funders had no role in study design, data collection and interpretation, or the decision to submit the work for publication.

### Author contributions

RVS, Contributed to the conception and design, Acquisition of data, Analyses and interpretation of data, Writing of software for simulations and the optimization of parameters, Drafting and revising the article; BK, Design and conduct of molecular dynamics simulations, Conception and design, Analysis and interpretation of data, Drafting or revising the article; KSS, DSR, FH, GE, DP, YL, Acquisition of data, Analysis and interpretation of data; YRH, Writing of software for simulations and the optimization of parameters, Analysis and interpretation of data; JVR, Provision of resources and expertise for microscropy and spectroscopy, Analysis and interpretation of data; CCG, Provision of resources and expertise for microscropy and spectroscopy, Conception and design, Analysis and interpretation of data, Drafting or revising the article; JE, Conception and design, Analysis and interpretation of data, Drafting or revising the article; JWW, Writing of software for simulations and the optimization of parameters, Conception and design, Analysis and interpretation of data, Drafting or revising the article

### Author ORCIDs

Rabindra V Shivnaraine, ⓘ http://orcid.org/0000-0002-1478-5422

## Additional files

**Supplementary files**

• Source code 1. Analysis of binding.

• Source code 2. Calculation of FRET by spectral unmixing.

• Source code 3. Simulation of the capped allosteric ternary complex model.

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

## Appendix 1: Extended Methods

### Fluorophore-tagged variants of the M₂ receptor

To maximize the quantum yield of FlAsH, the tetracysteine motif was bracketed by the flanking sequences FLN and MEP (*Martin et al., 2005*) (*i.e.*, FLNCCPGCCMEP, abbreviated FCM). The tripeptide FLN was added upstream of CCPGCC (forward primer, 5′-CTCTTCTGGCAGTTCATTGTATTCTTGAACTGTTGCCCAGGATG-3′), and MEP was added downstream of FLNCCPGCC (forward primer, 5′-TGCCCAGGATGTTGCATGGAGCCGGGGGTGAGAACTGTG-3′) to obtain the construct designated mCh-M₂-FCM.

To compare FlAsH with a fluorescent protein, mCherry was inserted in place of the FCM sequence after Val$^{166}$ in ECL2 of the eGFP-tagged receptor.

### Binding-defective mutants of the M₂ receptor

Negatively charged residues in the EDGE sequence at positions 172–175 of ECL2 were replaced by positively charged residues (*i.e.*, Lys$^{172}$Arg$^{173}$Gly$^{174}$Lys$^{175}$) (forward primer, 5′-CATTGTAGGGGTGAGAACTGTGAAGCGTGGGAAGTGCTACATTCAGTTTTTTTC-3′) in a receptor bearing hexahistidine at the *N*-terminus (*i.e.*, M₂-ECL2+ve). Aspartic acid at position 103 was replaced by alanine (forward primer, 5′-CTTTGGCTAGCCCTGGCCTATGTGGTCAGCAAT-3′) in a receptor bearing the FLAG epitope at the *N*-terminus [*i.e.*, M₂(D103A)] and in the FRET-based sensor [*i.e.*, mCh-M₂(D103A)-FCM].

### *Sf9* cells

The cells were cultured at 27 °C in Sf-900 II SFM insect cell medium supplemented with 2% fetal bovine serum, 1% Fungizone, and 50 µg/mL gentamycin (all from Gibco Life Technologies, Inc.). When growing at a density of $2 \times 10^6$ cells/mL, they were infected with one or both baculoviruses at a total multiplicity of infection of 5.

### CHO cells

Cells for transient transfections were grown in 5% $CO_2$ at 37°C in Dulbecco's Modified Eagle's Medium (DMEM, Gibco Life Technologies, Inc.) supplemented with 10% fetal calf serum, 1% non-essential amino acids, 100 U/mL penicillin, and 100 µg/mL streptomycin (all from Gibco Life Technologies, Inc.). Transfections were performed on cells growing at 50–75% confluence. Cells intended for microscopy were grown in 35 mm dishes containing a No. 1.5 glass coverslip (Mattek), which served as a 14 mm microwell, and were transfected with 1 µg total DNA using 3 µL GeneExpresso Max transfection reagent (Excellgen). Cells otherwise were grown in T-175 tissue-culture flasks, transfected with 120 µg total DNA using 360 µL GeneExpresso Max reagent, harvested by scraping in phosphate-buffered saline (PBS, 20 mM $KH_2PO_4$, 150 mM NaCl, adjusted to pH 7.40 with NaOH), centrifuged for 10 min at $3,000 \times g$ and 4 °C, and stored at −20 °C. In either case, two plasmids were transfected using equal amounts of DNA for each (*i.e.*, 0.5 µg or 60 µg). Frozen cells were thawed on ice as required and processed as described in Materials and Methods.

## Porcine atria

Briefly, fresh atria were washed twice with ice-cold PBS and homogenized in buffer B (20 mM imidazole, 1 mM EDTA, 0.1 mM PMSF, 0.02% $NaN_3$, adjusted to pH 7.60 with HCl) supplemented with benzamidine (1 mM), pepstatin A (20 µg/mL), leupeptin (0.2 µg/mL), and bacitracin (200 µg/mL). All constituents of PBS and buffer B were from Sigma-Aldrich. The resulting homogenate was fractionated by centrifugation on a sucrose density gradient (13–28%) to obtain the sarcolemmal fraction, which was collected by centrifugation, resuspended in buffer B, recovered by centrifugation, and stored at −70°C.

## Electrophoresis and western blotting

Samples were heated at 65°C for 5 min prior to loading on precast polyacrylamide gels from Bio-Rad (Ready Gel Tris-HCl, 10%). These conditions do not induce aggregation of the $M_2$ muscarinic receptor from *Sf9* cells. Resolved proteins were transferred onto nitrocellulose membranes (Bio-Rad, 0.45 µm), treated for 2 hr with the primary antibody (anti-$M_2$ muscarinic acetylcholine receptor monoclonal antibody, Thermoscientific) at a dilution of 1:1000, and then for 1 hr with the horseradish peroxidase-conjugated secondary antibody at a dilution of 1:3000. Proteins were visualized by chemiluminescence (ECL, Hyperfilm MP, GE Healthcare). The images were digitized at a resolution of 300 dpi, and the intensities of the bands were estimated from the densitometric trace using ImageJ (*Rasband, 1997*). Further details have been described previously (*Park et al., 2003*).

## Solution of equations, optimization of parameters, and estimation of means and standard errors of the mean

All equations were fitted to the data according to the Levenberg-Marquardt procedure unless indicated otherwise (*Marquardt, 1963*). *Equation 3* was solved numerically, as described previously (*Wells, 1992*); other equations were solved analytically. Equilibrium constants and potencies in *Equations 2 and 3* were optimized on a logarithmic scale; rate constants and other parameters were optimized throughout on a linear scale. The effects of various constraints on the weighted sum of squares were assessed by means of the *F*-statistic. Weighting of the data and other statistical procedures were performed as described previously (*Wells, 1992*). In the case of parametric values derived from one set of data or from two or more sets in a global analysis, the errors were estimated from the diagonal elements of the covariance matrix (*Equations 1–3*) or by bootstrapping (*Equations 6 and 9*). Parametric values that are the means of independent estimates are presented together with the standard error unless stated otherwise. Further details of the analyses and statistical procedures have been described previously (*Shivnaraine et al., 2012*; *Wells, 1992*).

## Appendix 2: Extended Details of the Molecular Modeling and Dynamics

### Modeling of unresolved regions at the C-terminus of mCherry, and the N-terminus and intracellular region of the receptor

Unresolved residues at the C-terminus of mCherry and the N-terminus of the $M_2$ receptor (i.e., mCh-GGMDELYKLE and NNSTNSSNNSLALTSPY-$M_2$) were modeled using a LowModeMD conformational search conducted in MOE with the CHARMM27 force field. The resulting structures were joined, and the new system was subjected to steepest-descent minimization and refined by a short molecular dynamics simulation (5 ns) in a distance-dependent dielectric implicit model of solvation. A restraining distance of 55.7 Å was maintained between FlAsH and the center of the mCherry barrel, in accordance with the experimentally measured distance calculated from the FRET efficiency and the Förster radius. The intracellular region of the receptor between transmembrane domains 5 and 6 (i.e., residues 233–374), which also is unresolved in 4MQS, was determined by homology modeling in Modeller 9.13 and protonated in MOE using the CHARMM27 force field at pH 7.0. The fusion product is shown in *Figure 2A*.

### Ligand–receptor and ligand–ligand interactions

Models of the receptor were based on crystal structures from which the G protein-mimetic nanobody (4MQS), the agonist iperoxo (M4QS), and the allosteric modulator LY2119620 (M4QT) were removed. Unresolved residues at the N- and C-termini were not included (i.e., residues 1–18 and 457–466). Part of the intracellular region between transmembrane domains 5 and 6 also is unresolved (residues 233–374); the loop therefore was shortened to RIKKDKKEPVANQDPVSTRKK, which was modeled using Modeller 9.13. Five models of the loop were generated, with the rest of the protein constrained to its crystallographic conformation, and the structure with the lowest molpdf score was selected for subsequent simulations. The receptor, including the optimized third intracellular loop, then was protonated in MOE using the CHARMM27 force field at pH 7.0.

All orthosteric and allosteric ligands were built in MOE and protonated using the CHARMM27 force field at pH 7.0. A set of conformers was generated by means of LowModeMD, and force-field partial charges were calculated for each conformer. Those conformations whose energy exceeded the minimum by 9.0 kcal/mol or more were discarded.

