## [Decision Letter]

Thank you for submitting your work entitled "The Mechanistic Basis of Allosteric Modulation in a G Protein-coupled Receptor" for consideration by *eLife*. Your article has been reviewed by two peer reviewers, and the evaluation has been overseen by Werner Kühlbrandt as the Reviewing Editor and Richard Aldrich as the Senior Editor.

The following individual involved in review of your submission has agreed to reveal their identity: Terry Hébert (peer reviewer).

The reviewers have discussed the reviews with one another and the Reviewing Editor has drafted this decision to help you prepare a revised submission.

Summary:

The reviewers agree that this is a very solid article that follows on from previous work from your group demonstrating that receptor oligomerization is the basis of much of the ligand binding cooperativity in M_2_ muscarinic receptors. The manuscript combines a novel FRET approach based on spectral unmixing with molecular modelling and molecular dynamics simulations to understanding the nature of positive and negative cooperativity between orthosteric and allosteric sites in the receptor. As in previous work based on ligand binding in purified monomers or oligomers, you show that only negative cooperativity can be detected between the orthosteric and allosteric sites using a FlAsH-FRET approach. Using binding defective mutants you show that cooperativity between these two sites is lost but can be restored by an untagged WT receptor, suggesting that this allostery is rescued only in the context of an oligomeric receptor. The approach is elegant and the data is supported by your previous binding studies. The reviewers point out that your group has consistently provided the best proof for receptor oligomerization and its allosteric consequences. Nevertheless, they had a few substantive concerns they would like you to address.

Essential revisions:

1) The title exaggerates the results dramatically and must be changed. The authors do not offer any molecular mechanism for how allosterism in an oligomer is propagated between receptors. The impact statement would make a far more accurate title.

2) The authors claim they have validated the functionality of their modified receptors. This is critical for the measures of allosteric interactions to have any meaning and to be interpreted by the MD simulations. In fact, they only show that the core FlAsH-mCherry construct binds ligand in detergent. They claim that they can localize FlAsH binding to the cell surface but do not show this anywhere (not even in one of the copious supplemental figures). In the overexpression context, some receptor must be inside the cell so it would be good to see where the receptor is and when it binds FlAsH. They also do not show function in a cellular context. Interfering with ECL2 is potentially disruptive. In any case, this would need to be done for *all* their constructs (even the ones that don't bind) in order for us to interpret their results accurately.

3) The authors claim that the anisotropy measurements demonstrate that the ECL2 moiety moves rather than the large GFP adduct. They show that is true under basal conditions, but what about in response to ligand? This is the only way to conclude that is doesn't move. Again, do these constructs need to be functional to interpret the data?

4) How do the in-cell experiments prove allostery? Why can't this be molecular crosstalk? The authors should formally exclude this with a second receptor that does not dimerize with the M_2_ receptor – or better yet measure the allostery in detergent-solubilized preps where crosstalk is not possible. They seem uniquely poised to do this.

5)In the first paragraph of the subsection “A FRET-based sensor of changes at the allosteric site” the authors mention that the affinity of the FRET-tagged receptor for NMS is unchanged, which is reassuring, but no data are shown for whether the extensively modified receptor used in the FRET experiments behaves in an identical fashion allosterically compared to the native receptor. This will require the purification of the tagged receptor, reconstitution into nanodiscs and liposomes and the experiments depicted in Figure 1 repeated.

6)In the sixth paragraph of the subsection “A FRET-based sensor of changes at the allosteric site”. The FRET efficiency was measured upon adding NMS alone, and then upon addition of NMS and gallamine together. The authors conclude that the change in FRET efficiency is due to the negative allosteric effect of gallamine, but do not do the control experiment of gallamine alone. This should be done.

7) Figure 2. A full statistical analysis needs to be performed on the data in Figure 2 and Figure 3 to define the significance of the differences observed in each case.

---

## [Author Response]

1) The title exaggerates the results dramatically and must be changed. The authors do not offer any molecular mechanism for how allosterism in an oligomer is propagated between receptors. The impact statement would make a far more accurate title.

Although the original title referred only to ‘mechanistic basis,’ not ‘molecular mechanism,’ we have provided a new title that avoids such terms altogether (i.e., ‘Allosteric modulation in monomers and oligomers of a G protein-coupled receptor’).

2) The authors claim they have validated the functionality of their modified receptors. This is critical for the measures of allosteric interactions to have any meaning and to be interpreted by the MD simulations. In fact, they only show that the core FlAsH-mCherry construct binds ligand in detergent. They claim that they can localize FlAsH binding to the cell surface but do not show this anywhere (not even in one of the copious supplemental figures). In the overexpression context, some receptor must be inside the cell so it would be good to see where the receptor is and when it binds FlAsH. They also do not show function in a cellular context. Interfering with ECL2 is potentially disruptive. In any case, this would need to be done for all their constructs (even the ones that don't bind) in order for us to interpret their results accurately.

Functionality of modified receptors. The original manuscript contained some information on the functionality of our modified receptors (points i and ii below), some of which was noted by the reviewers. We now have added additional data which we believe address their comment (points iii and iv).

i) We and others have shown previously that the functionality of muscarinic receptors is not affected by the fusion of a fluorophore at the *N*-terminus of muscarinic receptors [e.g., M_1_, Weill et al. (1999) J. Neurochem. 73, 791–801; M_2_, Pisterzi et al. (2010) J. Biol. Chem. 285, 16723–16738]. This was noted originally in the Materials and methods, and it now appears in the Results (subsection “A FRET-based sensor of allosteric effects”, second paragraph).

ii) Two constructs contained modifications in the second extracellular loop (ECL2). In one case (i.e., M_2_-ECL2+), the negatively charged groups within the EDGE sequence (positions 172–175 in ECL2) were replaced by positively charged ones (i.e., KRGK) to reduce the affinity for gallamine at the allosteric site while retaining that for ligands at the orthosteric site. This mutant was co-expressed with a mutant possessing a defective orthosteric site [i.e., M_2_(D103A)], and the purified heteromer exhibited allosteric effects of high and intermediate affinity for gallamine that resembled those observed previously with the wild-type receptor. These observations demonstrate that the mutant retains some native functionality, and they identify that functionality as a property of oligomers. The data are presented in Figure 3 of the original and revised manuscripts.

iii) The second construct contained a FlAsH-reactive insert in ECL2 and mCherry at the *N*-terminus (mCh-M_2_-FCM). We originally reported the affinity of solubilised mCh-M_2_-FCM for the antagonist *N*-[^3^H]methylscopolamine (NMS), as noted by the reviewers, but we did not assess its allosteric properties in binding assays. We now report that the binding of [^3^H]NMS to mCh-M_2_-FCM in CHO membranes and the modulation of that binding by the allosteric ligand gallamine are similar to what we reported previously for the wild-type receptor in CHO and myocardial membranes. The allosteric effect of gallamine is shown in Figure 2—figure supplement 2, and it demonstrates that the modified receptor retains the triphasic allosteric profile exhibited by the wild-type receptor. These results are introduced in the second paragraph of the subsection “A FRET-based sensor of allosteric effects.

iv) Localisation. We mentioned originally that FlAsH-reacted receptors were localised at the plasma membrane and that there was little or no emission from the cytosol. As the reviewers noted, however, there were no images. The revised manuscript includes confocal images of all of the constructs used in the investigation: namely, mCh-M_2_-FCM, mCh-M_2_(D103A)-FCM and M_2_-FCM (all reacted with FlAsH) as well as eGFP-M_2_, mCh-M_2_ and eGFP-M_2_-mCh. None of the modifications impaired the ability of the receptor to localise at the membrane, nor was there an appreciable presence of receptors in the cytosol. These results are presented in the second paragraph of the subsection “A FRET-based sensor of allosteric effects” and in Figure 2—figure supplement 1 of the revised manuscript.

3) The authors claim that the anisotropy measurements demonstrate that the ECL2 moiety moves rather than the large GFP adduct. They show that is true under basal conditions, but what about in response to ligand? This is the only way to conclude that is doesn't move. Again, do these constructs need to be functional to interpret the data?

The measurements of fluorescence anisotropy have been repeated for both of the constructs described in the original submission (i.e., eGFP-M_2_ and eGFP-truncM_2_). In addition, eGFP-M_2_ was examined in the presence of NMS, carbachol, gallamine and NMS + gallamine, as suggested by the reviewers, and we have performed our own measurements on eGFP alone. Finally, the rotational correlation times now are accompanied by the fluorescence lifetimes. These results are summarised in [Supplementary-material SD4-data], Panel A.

Measurements of the liganded state of the receptor were performed only on the full-length construct (i.e., eGFP-M_2_). As noted above (point no. 2), we have shown previously that the wild-type M_2_ receptor remains functional when a fluorophore is fused at the *N*-terminus. The truncated receptor was designed to probe for any dependence of the correlation time on the length of the link to the fluorophore. It was not examined for its allosteric properties, and it therefore seems unnecessary to measure its anisotropy in the liganded state.

Conclusions drawn from the rotational correlation times are the same as before, but the details are somewhat different from those described previously. In particular, the rapidly decaying (*φ* < 1 ns), minor component of the signal is no longer observed, and the anisotropy can be described by one exponential rather than two. The difference is due in part to a longer averaging time, giving a higher signal-to-noise ratio, and in part to improvements in the method of analysis. The fluorescence decay traces from the horizontal and vertical channels now are fit simultaneously upon deconvolution of the instrument response function assuming a mono-exponential decay of the intensity and another exponential term that defines the rotational correlation time. This leads to concurrent estimates of the fluorescence lifetime and the rotational correlation lifetime, and it provides a basis for distinguishing between free eGFP on the one hand and the eGFP-tagged receptors on the other.

4) How do the in-cell experiments prove allostery? Why can't this be molecular crosstalk? The authors should formally exclude this with a second receptor that does not dimerize with the M_2_ receptor – or better yet measure the allostery in detergent-solubilized preps where crosstalk is not possible. They seem uniquely poised to do this.

We believe that the results of the in-cell experiments argue persuasively for allostery between interacting protomers, as do those of the binding assays. Whether or not they arise from molecular cross-talk depends in part on one’s understanding of that term.

In our experiments, the intermolecular component of allostery has been confirmed directly – i.e., at the level of the receptor – using binding assays and FRET. The wild-type receptor was co-expressed with a mutant of the FRET sensor lacking the orthosteric site to demonstrate that binding of NMS to the former increased the FRET efficiency at the latter (Figure 3). The effect was repeated using an eGFP-tagged wild-type receptor and selecting only those cells that co-expressed both fluorophores i.e., eGFP on the wild-type receptor and mCherry on FlAsH-reacted mCh-M_2_(D103A)-FCM (Figure 3—figure supplement 1; subsection “Intermolecular modulation of FRET”, last paragraph). We also purified a heteromer of two mutants, one lacking the orthosteric site [i.e., FLAG-M_2_(D103A)] and one lacking the allosteric site (i.e., His6-M_2_-ECL2+ve; see also point no. 2iii above), and demonstrated that high-affinity effects of gallamine on the binding of [^3^H]NMS are a manifestation of interactions between the allosteric site of one protomer and the orthosteric site of another (Figure 3). These data point directly to allosteric interactions between two or more molecules of receptor.

Cross-talk often is used to describe the downstream convergent effects of two proteins that regulate or otherwise participate in the same signalling pathway. Related to this idea is the notion of a transient interaction between one protein and a third party that affects its interaction with a second protein. Finally, one can imagine something akin to a scaffold that would mediate between two proteins. The reviewers’ reference to studies in a solubilised preparation suggests to us that they are referring to the first or second of these interpretations. Both possibilities are difficult to reconcile with the results from any one of the experiments described above, and they seem highly unlikely in the light of all three. Also, the third experiment described above is in effect one of those recommended by the reviewers.

The notion of an unidentified third party acting as an intermediary in a stable ternary complex is more difficult to reject categorically, inasmuch as it could exist in the intact CHO cells (FRET) and survive purification (binding). We have studied purified and reconstituted M_2_ receptors at some length over the years, however, and we have no evidence that such a third party is present in oligomers or necessary for their existence. Given that such an arrangement seems speculative and unlikely, we have opted not to consider it in the present manuscript. This of course could be revisited if the reviewers think otherwise.

5) In the first paragraph of the subsection “A FRET-based sensor of changes at the allosteric site” the authors mention that the affinity of the FRET-tagged receptor for NMS is unchanged, which is reassuring, but no data are shown for whether the extensively modified receptor used in the FRET experiments behaves in an identical fashion allosterically compared to the native receptor. This will require the purification of the tagged receptor, reconstitution into nanodiscs and liposomes and the experiments depicted in Figure 1 repeated.

This comment seems to concern the same issue as that in point no. 2, which we have addressed experimentally as described above (point no. 2iii). In addition, however, the reviewers suggest that the experiments of Figure 1 be repeated on the mCherry- and FCM-modified receptor after its reconstitution in nanodiscs and liposomes. The reviewers evidently were thinking of our recent paper in which we compared the M_2_ receptor reconstituted as a monomer in nanodiscs and as a reassembled tetramer in liposomes [Redka et al. (2014)J. Biol. Chem. 289, 24347–24365].

We have not undertaken such experiments, which in our view would not warrant the very substantial commitment of time and resources. A baculovirus would have to be prepared, large quantities of the monomeric protein would have to be purified from *Sf*9 cells, the purified monomers would have to be reconstituted as monomers in nanodiscs and as tetramers in liposomes, and the two preparations would have to be characterised for their binding properties. The oligomeric state of the receptor would have to be confirmed at each step. All of this would take several months, which would put us well beyond the period allowed for revisions, and it would increase substantially the length of an already lengthy manuscript. It would be, in effect, a new investigation.

The reviewers have suggested that such an investigation is required to confirm that the allosteric properties of the wild-type receptor are not affected by the modification. We believe, however, that this is demonstrated adequately by the present data. As noted by the reviewers, we reported in the original submission that the affinity of [^3^H]NMS for the wild-type and modified receptors was the same. We now have shown that, in membranes from CHO cells, the modified receptor displays the characteristic triphasic effect of gallamine on the binding of [^3^H]NMS. We see that as persuasive evidence for the retention of the allosteric properties.

A further consideration here is that, in our view, recovery of the triphasic profile in reconstituted tetramers would do little to validate the nature of the receptor in CHO membranes; rather, it would demonstrate that reconstituted tetramers of the modified receptor have recovered the native allosteric properties. That would be of interest if one were planning experiments on reconstituted tetramers, and it certainly would support our claim that reconstituted tetramers are a faithful reconstruction of the native state [Ma et al. (2007) Biochemistry 46, 7907–7927; Redk aet al. (2013) Biochemistry 52, 7405–7427; Redka et al. (2014) ibid.], but those issues are beyond the scope of the present manuscript. If the native allosteric properties were not recovered upon reconstitution of the mutant in liposomes, it would raise questions about the nature of that reconstituted preparation, but it would not invalidate our conclusions regarding the modified receptor in CHO membranes or whole cells.

Finally, there is the question of reconstituting monomers in nanodiscs. This also would be a time-consuming exercise that seems to offer little if any gain. It is clear from the present data that all of the complexity represented by the triphasic, serpentine binding curves is absent from monomers in solution, which behave as one would expect for a single orthosteric site and a single allosteric site on the same protomer. When we compare affinities, it is between the single affinity of gallamine or strychnine for monomers in solution and the affinities for oligomers in solution. It is not clear to us how the story would be advanced by measurements on monomers in nanodiscs. Also, the binding of agonists and antagonists to monomers is essentially the same in solution and after reconstitution in nanodiscs [e.g., Redka et al. (2014) ibid.]. It seems reasonable to suppose that the same would be true of intramolecular allosteric effects.

6) In the sixth paragraph of the subsection “A FRET-based sensor of changes at the allosteric site”. The FRET efficiency was measured upon adding NMS alone, and then upon addition of NMS and gallamine together. The authors conclude that the change in FRET efficiency is due to the negative allosteric effect of gallamine, but do not do the control experiment of gallamine alone. This should be done.

The assays with gallamine have been performed, and the results have been included in the revised manuscript (subsection “A FRET-based sensor of allosteric effects”, seventh paragraph; Figure 2 (now 2E), [Supplementary-material SD4-data], Panel F).

7) Figure 2. A full statistical analysis needs to be performed on the data in Figure 2 and Figure 3 to define the significance of the differences observed in each case.

Statistical details pertaining to the data in Figure 2 (now 2E) and 3D are now shown in [Supplementary-material SD4-data], Panel F and [Supplementary-material SD5-data].